# Gating is Weighting: Understanding Gated Linear Attention through In-context Learning

## Abstract

Linear attention methods provide a strong alternative to softmax attention as they allow for efficient recurrent decoding. Recent research has focused on enhancing standard linear attention by incorporating gating while retaining its computational benefits. Such Gated Linear Attention (GLA) architectures include highly competitive models such as Mamba and RWKV. In this work, we examine the in-context learning capabilities of the GLA model and make the following contributions. We show that a multilayer GLA can implement a general class of Weighted Preconditioned Gradient Descent (WPGD) algorithms with data-dependent weights. These weights are induced by the gating and allows the model to control the contribution of individual tokens to prediction. To further understand the mechanics of weighting, we introduce a novel data model with multitask prompts and characterize the optimization landscape of the problem of learning a WPGD algorithm. We identify mild conditions under which there is a unique (global) minimum up to scaling invariance, and the associated WPGD algorithm is unique as well. Finally, we translate these findings to explore the optimization landscape of GLA and shed light on how gating facilitates context-aware learning and when it is provably better than vanilla linear attention.

## 1 Introduction

The Transformer architecture (Vaswani, 2017) has become the de facto standard for language modeling tasks. The key component of the Transformer is the self-attention mechanism, which computes softmax-based similarities between all token pairs. Despite its success, the self-attention mechanism has quadratic complexity with respect to sequence length, making it computationally expensive for long sequences. To address this issue, a growing body of work has proposed near-linear time approaches to sequence modeling. The initial approaches included linear attention and state-space models, both achieving $O(1)$ inference complexity per generated token, thanks to their recurrent form. While these initial architectures typically do not match softmax-attention in performance, recent recurrent models such as Mamba (Gu & Dao, 2023; Dao & Gu, 2024), mLSTM (Beck et al., 2024), GLA Transformer (Yang et al., 2023), and RWKV-6 (Peng et al., 2024) achieve highly competitive results with the softmax Transformer. Notably, as highlighted in Yang et al. (2023), these architectures can be viewed as variants of *gated linear attention* (GLA), which incorporates a gating mechanism within the recurrence of linear attention.

Given a sequence of tokens $(z_i)_{i=1}^n \subset \mathbb{R}^d$ and associated query, key, and value embeddings $(q_i, k_i, v_i)_{i=1}^n \subset \mathbb{R}^d$, with $d$ being the embedding dimension, the GLA recurrence is given by

$$S_i = G_i \odot S_{i-1} + v_i k_i^\top, \quad \text{and} \quad o_i = S_i q_i. \tag{1}$$

Here, $S_i \in \mathbb{R}^{d \times d}$ represents the *2D state variable*, $o_i \in \mathbb{R}^d$ represents the $i$'th output token, and the *gating variable* $G_i := g(z_i) \in \mathbb{R}^{d \times d}$ is applied to the state through the Hadamard product $\odot$. When the gating is removed, the model reduces to causal linear attention (Katharopoulos et al., 2020).

The central objective of this work is to enhance the mathematical understanding of the GLA mechanism. In-context learning (ICL), one of the most remarkable features of modern sequence models, provides a powerful framework to achieve this aim. ICL refers to the ability of a sequence model to implicitly infer functional relationships from the demonstrations provided in its context window (Brown, 2020; Min et al., 2022). It is inherently related to the model's ability to emulate learning algorithms. Notably, ICL has been a major topic of empirical and theoretical interest in recent years. More specifically, a series of works have examined the approximation and optimization characteristics of linear attention, and have provably connected linear attention to the preconditioned gradient descent algorithm (Von Oswald et al., 2023; Ahn et al., 2024; Zhang et al., 2024). Given that the GLA recurrence in (1) has a richer design space, this leads us to ask:

> **Q:** *What are the ICL capabilities of the GLA mechanism? What learning algorithm does it emulate when presented with an ICL task?*

**Contributions:** The GLA recurrence in (1) enables the sequence model to weight past information in a data-dependent manner through the gating mechanism $(\boldsymbol{G}_i)_{i=1}^n$. Building on this observation, we demonstrate that GLA models can implement a *data-dependent Weighted Preconditioned Gradient Descent (WPGD)* algorithm. Specifically, a one-step version of this algorithm with scalar gating, where all entries of $\boldsymbol{G}_i$ are identical, is described by the prediction:

$$\hat{y} = \boldsymbol{x}^\top \boldsymbol{P} \boldsymbol{X}^\top (\boldsymbol{y} \odot \boldsymbol{\omega}). \tag{2}$$

Here, $\boldsymbol{X} \in \mathbb{R}^{n \times d}$ is the input feature matrix; $\boldsymbol{y} \in \mathbb{R}^n$ is the associated label vector; $\boldsymbol{x} \in \mathbb{R}^d$ represents the test/query input to predict; $\boldsymbol{P} \in \mathbb{R}^{d \times d}$ is the preconditioning matrix; and $\boldsymbol{\omega} \in \mathbb{R}^n$ weights the individual samples. When $\boldsymbol{\omega}$ is fixed, we drop "data-dependent" and simply refer to this algorithm as the WPGD algorithm. However, for GLA, $\boldsymbol{\omega} := \boldsymbol{\omega}(\boldsymbol{X}, \boldsymbol{y})$ depends on the data through recursive multiplication of the gating variables. Building on this formalism, we make the following specific contributions:

⬦ **ICL capabilities of GLA (§3):** Through constructive arguments, we demonstrate that a multilayer GLA model can implement data-dependent WPGD iterations, with weights induced by the gating function. This construction sheds light on the role of causal masking and the expressivity distinctions between scalar- and vector-valued gating functions.

⬦ **Landscape of 1-step WPGD (§4):** The GLA⇔WPGD connection motivates us to ask: *How does WPGD weigh demonstrations in terms of their relevance to the query?* To address this, we study the fundamental problem of learning an optimal WPGD algorithm: Given a tuple $(\boldsymbol{X}, \boldsymbol{y}, \boldsymbol{x}, y) \sim \mathcal{D}$, with $y$ being the label associated with the query, we investigate the population risk minimization:

$$\mathcal{L}_{\text{WPGD}}^\star := \min_{\boldsymbol{P}, \boldsymbol{\omega}} \mathcal{L}_{\text{WPGD}}(\boldsymbol{P}, \boldsymbol{\omega}) \quad \text{where} \quad \mathcal{L}_{\text{WPGD}}(\boldsymbol{P}, \boldsymbol{\omega}) = \mathbb{E}_{\mathcal{D}}\left[\left(y - \boldsymbol{x}^\top \boldsymbol{P} \boldsymbol{X}(\boldsymbol{\omega} \odot \boldsymbol{y})\right)^2\right]. \tag{3}$$

As our primary mathematical contribution, we characterize the loss landscape under a general multitask data setting, where the tasks associated with the demonstrations $(\boldsymbol{X}, \boldsymbol{y})$ have varying degrees of correlation to the target task $(\boldsymbol{x}, y)$. We carefully analyze this loss landscape and show that, under mild conditions, there is a unique (global) minimum $(\boldsymbol{P}, \boldsymbol{\omega})$ up to scaling invariance, and the associated WPGD algorithm is also unique.

⬦ **Loss landscape of 1-layer GLA (§5):** The landscape is highly intricate due to the recursively multiplied gating variables. We show that learning the optimal GLA layer can be connected to solving (3) with a constraint $\boldsymbol{\omega} \in \mathcal{C}$, where the restriction $\mathcal{C}$ is induced by the choice of gating function and input space. Solidifying this connection, we introduce a multitask prompt model under which we characterize the loss landscape of GLA and the influence of task correlations. Our analysis and experiments reveal insightful distinctions between linear attention, GLA with scalar gating, and GLA with vector-valued gating.

## 1.1 RELATED WORK

We discuss prior literature under two topics.

**Efficient sequence models.** Recent sequence model proposals – such as RetNet (Sun et al., 2023), Mamba (Gu & Dao, 2023), xLSTM (Beck et al., 2024), GLA Transformer (Yang et al., 2023), RWKV-6 (Peng et al., 2024) – admit efficient recurrent forms while being increasingly competitive with the transformer architecture with softmax-attention. However, we have a rather limited theoretical understanding of these architectures, especially, when it comes to their optimization landscape and ICL capabilities. Park et al. (2024); Grazzi et al. (2024) demonstrate that Mamba is effective in competitive with a transformer of similar size in various ICL tasks whereas Arora et al. (2024); Jelassi et al. (2024) establish theoretical and empirical shortcomings of recurrent models for solving recall tasks. It is worth mentioning that, GLA models also connect to state-space models and linear RNNs (De et al., 2024; Orvieto et al., 2023; Gu et al., 2021; Fu et al., 2022), as they could be viewed as time-varying SSMs (Dao & Gu, 2024; Sieber et al., 2024). Finally, GLA models are also closely related to implicit self-attention frameworks. For example, the work by Zimerman et al. (2024) on unified implicit attention highlights how models such as Mamba (Gu & Dao, 2023) and RWKV (Peng et al., 2023) can be viewed under a shared attention mechanism. Additionally, Zong et al. (2024) leverage gated cross-attention for robust multimodal fusion, demonstrating another practical application of gated mechanisms. Both approaches align with GLA's data-dependent gating, suggesting its potential for explainability and stable fusion tasks.

**Theory of in-context learning.** The theoretical aspects of ICL has been studied by a growing body of works during the past few years (Xie et al.; von Oswald et al., 2023; Gatmiry et al.; Li et al., 2023; Collins et al., 2024; Wu et al., 2023; Fu et al.; Lin & Lee, 2024; Akyürek et al., 2023; Zhang et al., 2023). A subset of these follow the setting of Garg et al. (2022) which investigates the ICL ability of transformers by focusing on prompts where each example is labeled by a task function from a specific function class, such as linear models. Akyürek et al. (2023) focuses on linear regression and provide a transformer construction that can perform a single step of GD based on in-context examples. Similarly,

Von Oswald et al. (2023) provide a construction of weights in linear attention-only transformers that can replicate GD steps for a linear regression task on in-context examples. Notably, they observe similarities between their constructed networks and those resulting from training on ICL prompts for linear regression tasks. Building on these, Zhang et al. (2024); Mahankali et al. (2023); Ahn et al. (2024) focus on the loss landscape of ICL for linear attention models. For a single-layer model trained on in-context prompts for random linear regression tasks, Mahankali et al. (2023); Ahn et al. (2024) show that the resulting model performs a single preconditioned GD step on in-context examples in a test prompt, aligning with the findings of Von Oswald et al. (2023). More recent work (Ding et al., 2023) analyzes the challenges of causal masking in causal language models (causalLM), showing that their suboptimal convergence dynamics closely resemble those of online gradient descent with non-decaying step sizes. Additionally, Li et al. (2024) analyzes the landscape of the H3 architecture, an SSM, under the same dataset model. They show that H3 can implement WPGD thanks to its convolutional/SSM filter. However, their WPGD theory is restricted to the trivial setting of equal weights, relying on the standard prompt model with IID examples and shared tasks. In contrast, we propose novel multitask datasets and prompt models where nontrivial weighting is provably optimal. This allows us to characterize the loss landscape of WPGD and explore advanced GLA models, linking them to data-dependent WPGD algorithms.

## 2 PROBLEM SETUP

*Notations.* $\mathbb{R}^d$ is the $d$-dimensional real space, with $\mathbb{R}^d_+$ and $\mathbb{R}^d_{++}$ as its positive and strictly positive orthants. $[n]$ denotes $\{1, \cdots, n\}$. Bold letters, e.g., $\boldsymbol{a}$ and $\boldsymbol{A}$, represent vectors and matrices. The identity matrix of size $n$ is $\boldsymbol{I}_n$. $\mathbf{1}$ and $\mathbf{0}$ denote the all-one and all-zero vectors or matrices of proper size. $\mathcal{N}(\boldsymbol{\mu}, \boldsymbol{\Sigma})$ is the Gaussian distribution with mean $\boldsymbol{\mu}$ and covariance $\boldsymbol{\Sigma}$. The symbol $\odot$ denotes the Hadamard product and $\oslash$ denotes Hadamard division. Given $\boldsymbol{a}_{i+1}, \cdots, \boldsymbol{a}_j \in \mathbb{R}^d$, we use $\boldsymbol{a}_{i:j}$ to denote $\boldsymbol{a}_{i+1} \odot \cdots \odot \boldsymbol{a}_j$ for $i < j$, and $\boldsymbol{a}_{i:i} = \mathbf{1}_d$ is the $d$-dimensional all ones vector.

The objective of this work is to develop a theoretical understanding of GLA through ICL. The optimization landscape of standard linear attention has been a topic of significant interest in the ICL literature (Ahn et al., 2024; Li et al., 2024). Following these works, we consider the input prompt

$$\boldsymbol{Z} = [\boldsymbol{z}_1 \cdots \boldsymbol{z}_n \boldsymbol{z}_{n+1}]^\top = \begin{bmatrix} \boldsymbol{x}_1 & \cdots & \boldsymbol{x}_n & \boldsymbol{x}_{n+1} \\ y_1 & \cdots & y_n & 0 \end{bmatrix}^\top \in \mathbb{R}^{(n+1)\times(d+1)}, \quad (4)$$

where tokens encode the input-label pairs $(\boldsymbol{x}_i, y_i)_{i=1}^n \subset \mathbb{R}^d \times \mathbb{R}$. We aim to enable ICL by training a sequence model $F \in \mathbb{R}^{(n+1)\times(d+1)} \to \mathbb{R}$ that predicts the label $y := y_{n+1}$ associated with the query $\boldsymbol{x} := \boldsymbol{x}_{n+1}$. This model will utilize the demonstrations $(\boldsymbol{x}_i, y_i)_{i=1}^n$ to infer the mapping between $\boldsymbol{x}$ and $y$. Assuming that the data is distributed as $(y, \boldsymbol{Z}) \sim \mathcal{D}$, the ICL objective is defined as

$$\mathcal{L}(F) = \mathbb{E}_{\mathcal{D}} \left[ (y - F(\boldsymbol{Z}))^2 \right]. \quad (5)$$

**Linear attention and shared-task distribution.** Central to our paper is the choice of the function class $F$. When $F$ is a linear attention model, the prediction $F(\boldsymbol{Z})$ takes the form $\hat{y} = \boldsymbol{z}_{n+1}^\top \boldsymbol{W}_q \boldsymbol{W}_k^\top \boldsymbol{Z}^\top \boldsymbol{Z} \boldsymbol{W}_v \boldsymbol{h}$ where $\boldsymbol{W}_k, \boldsymbol{W}_q, \boldsymbol{W}_v \in \mathbb{R}^{(d+1)\times(d+1)}$ are attention parameters, and $\boldsymbol{h} \in \mathbb{R}^{d+1}$ is the linear prediction head. We assume that the in-context input-label pairs follow a *shared-task distribution*, where $\boldsymbol{\beta} \sim \mathcal{N}(0, \boldsymbol{\Sigma}_\beta)$, $\boldsymbol{x}_i$ are i.i.d. with $\boldsymbol{x}_i \sim \mathcal{N}(0, \boldsymbol{\Sigma}_x)$, and $y_i \sim \mathcal{N}(\boldsymbol{\beta}^\top \boldsymbol{x}_i, \sigma^2)$, where $\sigma \geq 0$ represents the noise level. Under this shared-task distribution, it is shown (Von Oswald et al., 2023; Ahn et al., 2024; Zhang et al., 2024) that the optimal one-layer linear attention predictor $\hat{\boldsymbol{\beta}}$ coincides with the one-step optimal preconditioned gradient descent. In particular, we have $\hat{\boldsymbol{\beta}} = \boldsymbol{P}^\star \boldsymbol{X}^\top \boldsymbol{y}$, where

$$\boldsymbol{P}^\star = \underset{\boldsymbol{P} \in \mathbb{R}^{d\times d}}{\operatorname{argmin}} \ \mathbb{E}_{\mathcal{D}} \left[ \left( y - \boldsymbol{x}^\top \boldsymbol{P} \boldsymbol{X}^\top \boldsymbol{y} \right)^2 \right] \quad \text{with} \quad \boldsymbol{X} := [\boldsymbol{x}_1 \cdots \boldsymbol{x}_n]^\top \quad \text{and} \quad \boldsymbol{y} := [y_1, \cdots, y_n]^\top. \quad (6)$$

**Linear attention and gating.** Given the input prompt $\boldsymbol{Z}$, let $\boldsymbol{Q} = \boldsymbol{Z}\boldsymbol{W}_q, \boldsymbol{K} = \boldsymbol{Z}\boldsymbol{W}_k$ and $\boldsymbol{V} = \boldsymbol{Z}\boldsymbol{W}_v$ be the corresponding query, key, and value embedding matrices, respectively. The output of *causal* linear attention at time $i$ can be computed in a recurrent form as $\boldsymbol{S}_i = \boldsymbol{S}_{i-1} + \boldsymbol{v}_i \boldsymbol{k}_i^\top$ and $\boldsymbol{o}_i = \boldsymbol{S}_i \boldsymbol{q}_i$ where $\boldsymbol{q}_i, \boldsymbol{k}_i, \boldsymbol{v}_i \in \mathbb{R}^{d+1}$ are the query, key, value embeddings of $\boldsymbol{z}_i$ and $\boldsymbol{S}_0 = \mathbf{0}$. This recurrent form implies that linear attention has $O(d^2)$ cost, that is independent of $N$, to generate per-token. As presented in (1), GLA follows the same structure as linear attention but with a gating mechanism, which equips the model with the option to pass or supress the history. As discussed in Yang et al. (2023), the different choices of the gating function correspond to different popular recurrent architectures such as Mamba (Gu & Dao, 2023), Mamba2 (Dao & Gu, 2024), RWKV (Peng et al., 2024), etc.

We will show that GLA can weigh the context window through gating, thus, its capabilities are linked to the WPGD algorithm described in (7). This will in turn facilitate GLA to effectively learn *multitask prompt distributions* described by $y_i \sim \mathcal{N}(\boldsymbol{\beta}_i^\top \boldsymbol{x}_i, \sigma^2)$ with $\boldsymbol{\beta}_i$'s not necessarily identical.

## 3  WHAT GRADIENT METHODS CAN GLA EMULATE?

In this section, we investigate the ICL capabilities of gated linear attention (GLA) and show that under suitable instantiations of model weights, GLA can implement *data-dependent* WPGD.

### 3.1  GLA AS A DATA-DEPENDENT WPGD PREDICTOR

**Data-Dependent WPGD.** Given $X$ and $y$ as defined in (6), consider the weighted least squares objective $\mathcal{L}(\boldsymbol{\beta}) = \sum_{i=1}^{n} \Omega_i \cdot (y_i - \boldsymbol{\beta}^\top \boldsymbol{x}_i)^2$ with weights $\boldsymbol{\Omega} \in \mathbb{R}^n$. To optimize this, we use gradient descent (GD) starting from zero initialization, $\boldsymbol{\beta}_0 = \mathbf{0}$ with a step size of $\eta = 1/2$. One step of standard GD is given by

$$\boldsymbol{\beta}_1 = \boldsymbol{\beta}_0 - \eta \nabla \mathcal{L}(\boldsymbol{\beta}_0) = \sum_{i=1}^{n} \Omega_i \cdot \boldsymbol{x}_i y_i = \boldsymbol{X}^\top (\boldsymbol{\Omega} \odot \boldsymbol{y}).$$

Given a test/query feature $\boldsymbol{x}$, the corresponding prediction is $\hat{y} = \boldsymbol{x}^\top \hat{\boldsymbol{\beta}}$ where $\hat{\boldsymbol{\beta}} = \boldsymbol{\beta}_1$. Additionally, if we were using *preconditioned* GD with a preconditioning/projection matrix $\boldsymbol{P} \in \mathbb{R}^{d \times d}$, one step iteration would take the form

$$\hat{y} = \boldsymbol{x}^\top \hat{\boldsymbol{\beta}}, \quad \text{where} \quad \hat{\boldsymbol{\beta}} = \boldsymbol{P}\boldsymbol{\beta}_1 = \boldsymbol{P}\boldsymbol{X}^\top (\boldsymbol{\Omega} \odot \boldsymbol{y}).$$

Above is the basic *scalar-weighted* WPGD predictor which weights individual datapoints. It turns out, *vector-valued gating* can facilitate a more general estimator which weights individual coordinates. To this aim, we introduce an extension as follows: Let $\boldsymbol{P}_1, \boldsymbol{P}_2 \in \mathbb{R}^{d \times d}$ denote the preconditioning matrices, and let $\boldsymbol{\Omega} \in \mathbb{R}^{n \times d}$ denote the *vector-valued weighting* matrix. Note that $\boldsymbol{\Omega}$ is now a matrix rather than vector to facilitate coordinate-wise weighting and will remain consistent throughout the paper. We can similarly define

$$\boldsymbol{\beta}_1^{\mathrm{gd}}(\boldsymbol{P}_1, \boldsymbol{P}_2, \boldsymbol{\Omega}) := \boldsymbol{P}_2 (\boldsymbol{X}\boldsymbol{P}_1 \odot \boldsymbol{\Omega})^\top \boldsymbol{y} \tag{7a}$$

as one-step of (generalized) WPGD. Its corresponding prediction on a test query $\boldsymbol{x}$ is:

$$\hat{y} = \boldsymbol{x}^\top \hat{\boldsymbol{\beta}}, \quad \text{where} \quad \hat{\boldsymbol{\beta}} = \boldsymbol{\beta}_1^{\mathrm{gd}}(\boldsymbol{P}_1, \boldsymbol{P}_2, \boldsymbol{\Omega}). \tag{7b}$$

We note that by removing the preconditioning matrices $\boldsymbol{P}_1$, $\boldsymbol{P}_2$, and the weighting matrix $\boldsymbol{\Omega}$ in (7a), it reduces to standard GD. We also note that Li et al. (2024) demonstrates that H3-like models implement one-step WPGD, where the weighting is example-wise, i.e., setting $\boldsymbol{\Omega} = \boldsymbol{\omega}\mathbf{1}_d^\top$, and they focus on the shared-task distribution where $\boldsymbol{\beta}_i \equiv \boldsymbol{\beta}$. In contrast, our work considers a more general data setting where tasks within an in-context prompt are not necessarily identical.

We first introduce the following model constructions under which we establish the equivalence between GLA (c.f. (1)) and WPGD (c.f. (7)) with the weighting matrix induced by the input data and the gating function. Inspired by previous works (Von Oswald et al., 2023; Ahn et al., 2024), we consider the following restricted attention matrices:

$$\boldsymbol{W}_k = \begin{bmatrix} \boldsymbol{P}_k^\top & \mathbf{0} \\ \mathbf{0} & 0 \end{bmatrix}, \quad \boldsymbol{W}_q = \begin{bmatrix} \boldsymbol{P}_q^\top & \mathbf{0} \\ \mathbf{0} & 0 \end{bmatrix} \quad \text{and} \quad \boldsymbol{W}_v = \begin{bmatrix} \mathbf{0}_{d \times d} & \mathbf{0} \\ \mathbf{0} & 1 \end{bmatrix}, \tag{8}$$

where $\boldsymbol{P}_k, \boldsymbol{P}_q \in \mathbb{R}^{d \times d}$. Here note that we set the $(d+1, d+1)$'th entry of $\boldsymbol{W}_v$ to be one for simplification. More generally, it can be any nonzero number, e.g., $v \in \mathbb{R}$. Then parameterizing $\boldsymbol{W}_q$ with $\boldsymbol{P}_q/v$ returns the same output as from (8).

**Theorem 1.** *Recall the GLA from* (1) *and input sequence* $\boldsymbol{Z}$ *from* (4)*, and suppose that at time $i$, gating function has the form of $g(\boldsymbol{z}_i) = \boldsymbol{G}_i \in \mathbb{R}^{(d+1) \times (d+1)}$. Considering model construction in* (8) *and prediction head $\boldsymbol{h} = \mathbf{1}$, the single-layer GLA prediction returns*

$$f_{\mathsf{GLA}}(\boldsymbol{Z}) := \boldsymbol{o}_{n+1}^\top \boldsymbol{h} = \hat{\boldsymbol{\beta}}^\top \boldsymbol{x} \quad \textit{where} \quad \hat{\boldsymbol{\beta}} = \boldsymbol{\beta}_1^{gd}(\boldsymbol{P}_k, \boldsymbol{P}_q, \boldsymbol{\Omega}).$$

*Here, $\boldsymbol{\beta}_1^{gd}(\cdot)$ is a one-step WPGD feature predictor defined in* (7a)*, $\boldsymbol{P}_k, \boldsymbol{P}_q$ correspond to attention weights following* (8)*, and $\boldsymbol{\Omega} = [\boldsymbol{g}_{1:n+1} \ \boldsymbol{g}_{2:n+1} \ \cdots \ \boldsymbol{g}_{n:n+1}]^\top \in \mathbb{R}^{n \times d}$ where $\boldsymbol{g}_{i:n+1}, i \in [n]$ is given by*

$$\boldsymbol{g}_{i:n+1} := (\boldsymbol{g}_{i+1} \odot \boldsymbol{g}_{i+2} \cdots \boldsymbol{g}_{n+1}) \in \mathbb{R}^d \quad \textit{and} \quad \boldsymbol{G}_i = \begin{bmatrix} * & * \\ \boldsymbol{g}_i^\top & * \end{bmatrix} \tag{9}$$

Here and throughout, we use $*$ to fill the entries of the matrices that do not affect the final output, and based on the model construction given in (8), these entries can be assigned any value.

Observe that, crucially, since $\boldsymbol{g}_i$ (or $\boldsymbol{G}_i$) is associated with $\boldsymbol{z}_i$, $\boldsymbol{z}_i$ influences the weighting of all history $\boldsymbol{z}_{j<i}$. We defer the proof of Theorem 1 to the Appendix B.1. It is noticeable that only $d$ of the total $(d+1)^2$ entries in each gating matrix $\boldsymbol{G}_i$ are useful due to the model construction presented in (8). However, if we relax the weight restriction, e.g., $\boldsymbol{W}_v = [\mathbf{0}_{(d+1) \times d} \ \mathbf{1}_{d+1}]$, then the weighting matrix $\boldsymbol{\Omega}$ in Theorem 1 is associated with all rows of the $\boldsymbol{G}_i$ matrices. We defer the discussion to Appendix B.1.

## 3.2 Capabilities of multi-layer GLA

Ahn et al. (2024) demonstrated that, with appropriate construction, an $L$-layer linear attention model performs $L$-step preconditioned gradient descent on the dataset $(\boldsymbol{x}_i, y_i)_{i=1}^n$ provided within the prompt. In this work, we study multi-layer GLA and analyze the associated algorithm class it can emulate. It is worth mentioning that Ahn et al. (2024) does not consider *causal masking* which is integral to multilayer GLA due to its recurrent nature described in (1). Our analysis will capture the impact of gating and causal mask through $n$ separate gradient descent trajectories that are coupled.

Consider an $L$-layer GLA model. For $\ell \in [L]$, let $\boldsymbol{Z}_\ell$ and $\boldsymbol{O}_\ell$ denote the input and output of the $\ell$'th layer. In practice, residual connections are commonly applied. Hence, we define the updated output of the $\ell$'th layer (after applying the residual connection) as $\tilde{\boldsymbol{O}}_\ell := \boldsymbol{Z}_\ell + \boldsymbol{O}_\ell$. Note that $\tilde{\boldsymbol{O}}_\ell$ also serves as the input to the $(\ell + 1)$'th layer, i.e., $\boldsymbol{Z}_{\ell+1} = \tilde{\boldsymbol{O}}_\ell$. In the following, we focus on $(d + 1)$'th entries of each token's output at each layer, denoted by $\tilde{o}_{i,\ell} := (\tilde{\boldsymbol{O}}_\ell)_{i,d+1}$ for $i \in [n + 1], \ell \in [L]$.

**Theorem 2.** *Consider an $L$-layer GLA with residual connections, where $\boldsymbol{W}_k$ and $\boldsymbol{W}_q$ in the $\ell$'th layer are parameterized by $\boldsymbol{P}_{k,\ell}, \boldsymbol{P}_{q,\ell} \in \mathbb{R}^{d \times d}$, following (8), for $\ell \in [L]$. Let the gating be a function of the features, e.g., $\boldsymbol{G}_i = g(\boldsymbol{x}_i)$, and let $\boldsymbol{\Omega}$ be defined as in Theorem 1. Additionally, denote the masking as $\boldsymbol{M}_i = \begin{bmatrix} \boldsymbol{I}_i & \boldsymbol{0} \\ \boldsymbol{0} & \boldsymbol{0} \end{bmatrix} \in \mathbb{R}^{n \times n}$, and let $\hat{\boldsymbol{\beta}}_0, \boldsymbol{\beta}_{i,0} = \boldsymbol{0}$ for $i \in [n]$.*

*Then the $(d + 1)$'th entry of the $i$'th token at the $\ell$'th layer outputs:*

- *For $i \leq n$, $\tilde{o}_{i,\ell} = y_i - \boldsymbol{x}_i^\top \boldsymbol{\beta}_{i,\ell}$ where $\boldsymbol{\beta}_{i,\ell} = \boldsymbol{\beta}_{i,\ell-1} + \boldsymbol{P}_{q,\ell} (\nabla_{i,\ell} \oslash \boldsymbol{g}_{i:n+1})$,*

- *$\tilde{o}_{n+1,\ell} = -\boldsymbol{x}^\top \hat{\boldsymbol{\beta}}_\ell$ where $\hat{\boldsymbol{\beta}}_\ell = (1 + \alpha_\ell)\hat{\boldsymbol{\beta}}_{\ell-1} + \boldsymbol{P}_{q,\ell} (\nabla_{n,\ell} \oslash \boldsymbol{g}_{n+1})$ and $\alpha_\ell = \boldsymbol{x}^\top \boldsymbol{P}_{q,\ell} \boldsymbol{P}_{k,\ell}^\top \boldsymbol{x}$.*

*Here, letting $\boldsymbol{B}_\ell = [\boldsymbol{\beta}_{1,\ell} \cdots \boldsymbol{\beta}_{n,\ell}]^\top$, $\bar{\boldsymbol{X}}_\ell = \boldsymbol{X}\boldsymbol{P}_{k,\ell} \odot \boldsymbol{\Omega}$, and $\hat{\boldsymbol{y}}_\ell = (\boldsymbol{X} \odot \boldsymbol{B}_{\ell-1})\boldsymbol{1}$, we define*

$$\nabla_{i,\ell} = \bar{\boldsymbol{X}}_\ell^\top \boldsymbol{M}_i (\hat{\boldsymbol{y}}_\ell - \boldsymbol{y}).$$

We defer the proof of Theorem 2 to the Appendix B.2. Theorem 2 states that an $L$-layer GLA implements $L$ steps of WPGD but with gradient in a recurrent form. To recap, given data $(\boldsymbol{X}, \boldsymbol{y})$ and prediction $\hat{\boldsymbol{\beta}}$, the gradient with respect to the squared loss takes the form $\boldsymbol{X}^\top(\boldsymbol{X}\hat{\boldsymbol{\beta}} - \boldsymbol{y})$, up to some constant $c$. In comparison, $\boldsymbol{P}_{q,\ell} (\nabla_{i,\ell} \oslash \boldsymbol{g}_{i:n+1})$ similarly acts as a gradient but incorporates layer-wise feature preconditioners ($\boldsymbol{P}_{q,\ell}, \boldsymbol{P}_{k,\ell}$), data weighting ($\boldsymbol{\Omega}$), and causality ($\boldsymbol{g}_{i:n+1}, \boldsymbol{M}_i$). Here, $\boldsymbol{M}_i$ represents causal masking, ensuring that at time $i$, only inputs from $j \leq i$ are used for prediction. Notably, the recurrent structure of GLA allows the gating mechanism to apply context-dependent weighting strategies. These results are consistent with Ding et al. (2023), which demonstrate that causal masking limits convergence by introducing sequence biases, akin to online gradient descent with non-decaying step sizes.

To simplify the theorem statement, we assume that the gating function depends only on the input feature, e.g., $\boldsymbol{G}_i = g(\boldsymbol{x}_i)$, ensuring that the corresponding data-dependent weighting is uniform across all layers. This assumption is included solely for clarity in the theorem statement, and the complete result is provided in Appendix B.2. Note that our inclusion of the additional term $\alpha_\ell$ captures the influence of the last token's output on the next layer's prediction, which is not addressed by Ahn et al. (2024). Based on the above multi-layer GLA result, we have the following corollary for multi-layer linear attention network with causal mask in each layer.

**Corollary 1.** *Consider an $L$-layer linear attention model with causal mask and residual connection in each layer. Let $\ell$'th layer be parameterized by $\boldsymbol{P}_{q,\ell}, \boldsymbol{P}_{k,\ell}$ as in (8) and define $\boldsymbol{P}_\ell := \boldsymbol{P}_{q,\ell} \boldsymbol{P}_{k,\ell}^\top$, $\ell \in [L]$. Let $\hat{\boldsymbol{\beta}}_0, \boldsymbol{\beta}_{i,0} = \boldsymbol{0}$ for $i \in [n]$. Then, the $(d + 1)$'th entry of the $i$'th token of the $\ell$'th layer outputs satisfies:*

- *For $i \leq n$, $\tilde{o}_{i,\ell} = y_i - \boldsymbol{x}_i^\top \boldsymbol{\beta}_{i,\ell}$ where $\boldsymbol{\beta}_{i,\ell} = \boldsymbol{\beta}_{i,\ell-1} + \boldsymbol{P}_\ell \nabla_{i,\ell}$,*

- *$\tilde{o}_{n+1,\ell} = -\boldsymbol{x}^\top \hat{\boldsymbol{\beta}}_\ell$ where $\hat{\boldsymbol{\beta}}_\ell = (1 + \alpha_\ell)\hat{\boldsymbol{\beta}}_{\ell-1} + \boldsymbol{P}_\ell \nabla_{n,\ell}$ and $\alpha_\ell = \boldsymbol{x}^\top \boldsymbol{P}_\ell \boldsymbol{x}$.*

*Here, we define $\nabla_{i,\ell} = \boldsymbol{X}^\top \boldsymbol{M}_i (\hat{\boldsymbol{y}}_\ell - \boldsymbol{y})$ with $\hat{\boldsymbol{y}}_\ell, \boldsymbol{M}_i$ following the same definitions as in Theorem 2.*

Our theoretical results in Theorem 2 focus on multi-layer GLA without Multi-Layer Perceptron (MLP) layers to isolate and analyze the effects of the gating mechanism. However, MLP layers, a key component of standard Transformers, facilitate further nonlinear feature transformations and interactions, potentially enhancing GLA's expressive power. Future work could explore the theoretical foundations of integrating MLPs into GLA and analyze the optimization landscape of general gated attention models, aligning them more closely with conventional Transformer architectures (Gu & Dao, 2023; Dao & Gu, 2024; Peng et al., 2024).

### 3.3 GLA with scalar gating

Theorem 1 establishes a connection between 1-layer GLA (c.f. (1)) and one-step WPGD (c.f. (7)), where the weighting in WPGD corresponds to the gating $g(z_i) = G_i$ in GLA, as detailed in Theorem 1. Now let us consider the widely used types of gating functions, such as $G_i = \alpha_i \mathbf{1}_{d+1}^\top$ (Yang et al., 2023; Katsch, 2023; Qin et al., 2024; Peng et al., 2024) or $G_i = \gamma_i \mathbf{1}_{d+1} \mathbf{1}_{d+1}^\top$ (Dao & Gu, 2024; Beck et al., 2024; Peng et al., 2021; Sun et al., 2024) where $\alpha_i \in \mathbb{R}^{d+1}$ and $\gamma_i \in \mathbb{R}$. In both cases, the gating matrices in (9) take the form of $\begin{bmatrix} * & * \\ g_i \mathbf{1}_d^\top & * \end{bmatrix}$, thus simplifying the predictor to a sample-weighted PGD, as given by

$$f_{\text{GLA}}(\mathbf{Z}) = \hat{\boldsymbol{\beta}}^\top \mathbf{x}, \quad \text{with} \quad \hat{\boldsymbol{\beta}} = \mathbf{P}\mathbf{X}^\top(\boldsymbol{\omega} \odot \mathbf{y}), \tag{10}$$

where $\mathbf{P} = \mathbf{P}_q \mathbf{P}_k^\top$ and $\boldsymbol{\omega} = [g_{1:n+1} \cdots g_{n:n+1}]^\top \in \mathbb{R}^n$. In the remainder, we will mostly focus on the 1-layer GLA with scalar gating as presented in (10).

## 4 Optimization landscape of WPGD

In this section, we explore the problem of learning the optimal sample-weighted PGD algorithm described in (10), a key step leading to our analysis of GLA. The problem is as follows. Recap from (6) that we are given the tuple $(\mathbf{x}, y, \mathbf{X}, \mathbf{y}) \sim \mathcal{D}$, where $\mathbf{X} \in \mathbb{R}^{n \times d}$ is the input matrix, $\mathbf{y} \in \mathbb{R}^n$ is the label vector, $\mathbf{x} \in \mathbb{R}^d$ is the query, and $y \in \mathbb{R}$ is its associated label. The goal is to use $\mathbf{X}, \mathbf{y}$ to predict $y$ given $\mathbf{x}$ via the 1-step WPGD prediction $\hat{y} = \mathbf{x}^\top \hat{\boldsymbol{\beta}}$, with $\hat{\boldsymbol{\beta}}$ as in (10). The algorithm learning problem is given by (3) which minimizes the WPGD risk $\mathbb{E}_{\mathcal{D}}[(y - \mathbf{x}^\top \mathbf{P}\mathbf{X}(\boldsymbol{\omega} \odot \mathbf{y}))^2]$.

Prior research (Mahankali et al., 2023; Li et al., 2024; Ahn et al., 2024) has studied the problem of learning PGD when input-label pairs follow an IID distribution. It is worth noting that while Li et al. (2024) establishes a connection between H3-like models and (10) similar to ours, their work assumes that the optimal $\boldsymbol{\omega}$ consists of all ones and does not specifically explore the optimization landscape of $\boldsymbol{\omega}$ when in-context samples are non-IID. Departing from this, we introduce a realistic model where each input-label pair is allowed to come from a distinct task.

**Definition 1** (Correlated task model). *Suppose $\boldsymbol{\beta}_i \in \mathbb{R}^d \sim \mathcal{N}(0, \mathbf{I})$ are jointly Gaussian for $i \in [n+1]$. Define the pairwise correlations $r_{ij} = \mathbb{E}[\boldsymbol{\beta}_i^\top \boldsymbol{\beta}_j]/d$ for $i, j \in [n+1]$, and the task and correlation matrices*

$$\boldsymbol{\beta} := \boldsymbol{\beta}_{n+1}, \quad \mathbf{B} = [\boldsymbol{\beta}_1 \ldots \boldsymbol{\beta}_n]^\top, \quad \mathbf{R} = \frac{1}{d}\mathbb{E}[\mathbf{B}\mathbf{B}^\top], \quad \text{and} \quad \mathbf{r} = \frac{1}{d}\mathbb{E}[\mathbf{B}\boldsymbol{\beta}]. \tag{11}$$

*Additionally, for any $i, j \in [n+1]$, $\boldsymbol{\beta}_i - r_{ij}\boldsymbol{\beta}_j$ is independent of $\boldsymbol{\beta}_j$.*

Note that in (11), we have $\mathbf{B} \in \mathbb{R}^{n \times d}$, $\mathbf{R} \in \mathbb{R}^{n \times n}$, and $\mathbf{r} \in \mathbb{R}^n$, with normalization ensuring that the entries of $\mathbf{R}$ and $\mathbf{r}$ lie in the range $[-1, 1]$, corresponding to correlation coefficients.

**Definition 2** (Multitask distribution). *$(\boldsymbol{\beta}_i)_{i=1}^{n+1}$ are drawn according to the correlated task model of Definition 1, $(\mathbf{x}_i)_{i=1}^{n+1} \in \mathbb{R}^d$ are IID following $\mathbf{x}_i \sim \mathcal{N}(0, \boldsymbol{\Sigma})$ and $y_i \sim \mathcal{N}(\mathbf{x}_i^\top \boldsymbol{\beta}_i, \sigma^2)$ for $i \in [n+1]$.*

**Definition 3.** *Let the eigen decompositions of $\boldsymbol{\Sigma}$ and $\mathbf{R}$ be denoted by $\boldsymbol{\Sigma} = \mathbf{U}\text{diag}(\mathbf{s})\mathbf{U}^\top$ and $\mathbf{R} = \mathbf{E}\text{diag}(\boldsymbol{\lambda})\mathbf{E}^\top$, where $\mathbf{s} = [s_1, \ldots, s_d]^\top \in \mathbb{R}_{++}^d$ and $\boldsymbol{\lambda} = [\lambda_1, \ldots, \lambda_n]^\top \in \mathbb{R}_+^n$. Let $s_{\min}$ and $s_{\max}$ denote the smallest and largest eigenvalues of $\boldsymbol{\Sigma}$, respectively. Further, let $\lambda_{\min}$ and $\lambda_{\max}$ denote the nonzero smallest and largest eigenvalues of $\mathbf{R}$. Define the effective spectral gap of $\boldsymbol{\Sigma}$ and $\mathbf{R}$, respectively, as*

$$\Delta_{\boldsymbol{\Sigma}} := s_{\max} - s_{\min}, \quad \text{and} \quad \Delta_{\mathbf{R}} := \lambda_{\max} - \lambda_{\min}. \tag{12}$$

**Assumption A.** *For the correlation vector $\mathbf{r}$ from (11), we have $\mathbf{r} = \mathbf{E}\mathbf{a}$ for some $\mathbf{a} = [a_1, \ldots, a_n]^\top \in \mathbb{R}^n$ with at least one nonzero $a_i$.*

Assumption A essentially ensures that $\mathbf{r}$ (representing the correlations between in-context tasks) can be expressed as a linear transformation of a vector $\mathbf{a}$ of nonzero values. This guarantees that the correlation structure is non-degenerate, meaning that all elements of $\mathbf{r}$ are influenced by meaningful correlations. Assumption A avoids trivial cases where there are no correlations between tasks. By requiring at least one nonzero element in $\mathbf{a}$, the assumption ensures that the tasks are interrelated.

The following theorem characterizes the stationary points $(\mathbf{P}, \boldsymbol{\omega})$ of the WPGD objective in (3).

**Theorem 3.** *Consider independent linear data as described in Definition 2. Suppose Assumption A on the correlation vector $\boldsymbol{r}$ holds. Let the functions $h : \mathbb{R}_+ \to \mathbb{R}_+$ and $g : \mathbb{R}_+ \to \mathbb{R}_+$ be defined as*

$$h(\bar{\gamma}) := \sum_{i=1}^{n} \frac{\lambda_i a_i^2}{(1 + \lambda_i \bar{\gamma})^2} \left( \sum_{i=1}^{n} \frac{a_i^2}{(1 + \lambda_i \bar{\gamma})^2} \right)^{-1}, \tag{13a}$$

$$g(\gamma) := \left( 1 + M \sum_{i=1}^{d} \frac{s_i^2}{(M + s_i(\gamma + 1))^2} \left( \sum_{i=1}^{d} \frac{s_i^3}{(M + s_i(\gamma + 1))^2} \right)^{-1} \right)^{-1}, \tag{13b}$$

*where $\{s_i\}_{i=1}^{d}$ and $\{\lambda_i\}_{i=1}^{n}$ are the eigenvalues of $\boldsymbol{\Sigma}$ and $\boldsymbol{R}$, respectively; $\{a_i\}_{i=1}^{n}$ are as given in Assumption A; and $M = \sigma^2 + \sum_{i=1}^{d} s_i$.*

*The risk function $\mathcal{L}(\boldsymbol{P}, \boldsymbol{\omega})$ in (3) has a stationary point $(\boldsymbol{P}^\star, \boldsymbol{\omega}^\star)$, up to rescaling, defined as*

$$\boldsymbol{P}^\star = \boldsymbol{\Sigma}^{-\frac{1}{2}} \left( \frac{\gamma^\star + 1}{\sigma^2 + \mathtt{tr}(\boldsymbol{\Sigma})} \cdot \boldsymbol{\Sigma} + \boldsymbol{I} \right)^{-1} \boldsymbol{\Sigma}^{-\frac{1}{2}}, \quad \text{and} \quad \boldsymbol{\omega}^\star = \left( g(\gamma^\star) \cdot \boldsymbol{R} + \boldsymbol{I} \right)^{-1} \boldsymbol{r}, \tag{14}$$

*where $\gamma^\star$ is a fixed point of composite function $h(g(\gamma))$.*

Theorem 3 characterizes the stationary points $(\boldsymbol{P}^\star, \boldsymbol{\omega}^\star)$, which exist up to re-scaling. This result presents the first landscape analysis of GLA for the joint learning of $(\boldsymbol{P}, \boldsymbol{\omega})$, while also exploring the stationary points $(\boldsymbol{P}^\star, \boldsymbol{\omega}^\star)$. In the following, we provide mild conditions on effective spectral gaps of $\boldsymbol{R}$ and $\boldsymbol{\Sigma}$ under which a unique (global) minimum $(\boldsymbol{P}^\star, \boldsymbol{\omega}^\star)$ exists.

**Theorem 4** (Uniqueness of the WPGD Predictor). *Consider independent linear data as given in Definition 2. Suppose Assumption A on the correlation vector $\boldsymbol{r}$ holds, and*

$$\Delta_{\boldsymbol{\Sigma}} \cdot \Delta_{\boldsymbol{R}} < M + s_{\min}, \tag{15}$$

*where $\Delta_{\boldsymbol{\Sigma}}$ and $\Delta_{\boldsymbol{R}}$ denote the effective spectral gaps of $\boldsymbol{\Sigma}$ and $\boldsymbol{R}$, respectively, as given in (12); $s_{\min}$ is the smallest eigenvalue of $\boldsymbol{\Sigma}$; and $M = \sigma^2 + \sum_{i=1}^{d} s_i$.*

**T1** *The composite function $h(g(\gamma))$ is a contraction mapping and admits a unique fixed point $\gamma = \gamma^\star$.*

**T2** *The function $\mathcal{L}(\boldsymbol{P}, \boldsymbol{\omega})$ has a unique (global) minima $(\boldsymbol{P}^\star, \boldsymbol{\omega}^\star)$, up to re-scaling, given by (14).*

*Proof Sketch.* Let $\gamma := \frac{\boldsymbol{\omega}^\top \boldsymbol{R} \boldsymbol{\omega}}{\|\boldsymbol{\omega}\|^2}$. Note that $\gamma \geq 0$ since $\boldsymbol{R}$ is positive semi-definite. From the first-order optimality condition, the solution to (3) takes the following form:

$$\boldsymbol{P}(\gamma) = C(\boldsymbol{r}, \boldsymbol{\omega}, \boldsymbol{\Sigma}) \cdot \boldsymbol{\Sigma}^{-\frac{1}{2}} \left( \frac{\gamma + 1}{\sigma^2 + \mathtt{tr}(\boldsymbol{\Sigma})} \cdot \boldsymbol{\Sigma} + \boldsymbol{I} \right)^{-1} \boldsymbol{\Sigma}^{-\frac{1}{2}}, \tag{16a}$$

$$\boldsymbol{\omega}(\gamma) = c(\boldsymbol{r}, \boldsymbol{\omega}, \boldsymbol{\Sigma}) \cdot \left( g(\gamma) \cdot \boldsymbol{R} + \boldsymbol{I} \right)^{-1} \boldsymbol{r}, \tag{16b}$$

for some constants $C(\boldsymbol{r}, \boldsymbol{\omega}, \boldsymbol{\Sigma})$ and $c(\boldsymbol{r}, \boldsymbol{\omega}, \boldsymbol{\Sigma})$.

Substituting the expression for $\boldsymbol{\omega}(\gamma)$ into $\gamma = \frac{\boldsymbol{\omega}^\top \boldsymbol{R} \boldsymbol{\omega}}{\|\boldsymbol{\omega}\|^2}$, and applying Assumption A, we obtain the equation $\gamma = h(g(\gamma))$. We then show that whenever $\Delta_{\boldsymbol{\Sigma}} \cdot \Delta_{\boldsymbol{R}} < M + s_{\min}$, the mapping $h(g(\gamma))$ is a contraction (see Lemma 1). By the Banach Fixed-Point Theorem, this guarantees the existence of a unique fixed point $\gamma = \gamma^\star$, where $\gamma^\star = h(g(\gamma^\star))$. Finally, substituting $\gamma^\star$ into (16) implies that $(\boldsymbol{P}^\star, \boldsymbol{\omega}^\star)$, as given in (14), is a unique (global) minima of (3), up to re-scaling. See Appendix C.2 for the complete proof of Theorem 4. □

Theorem 4 establishes mild conditions under which a unique (global) minimum $(\boldsymbol{P}^\star, \boldsymbol{\omega}^\star)$ exists, up to scaling invariance, and guarantees the uniqueness of the associated WPGD algorithm. It provides the first global landscape analysis for GLA and generalizes prior work (Li et al., 2024; Ahn et al., 2024) on the global landscape by extending the optimization properties of linear attention to the more complex *nonconvex* GLA with joint $(\boldsymbol{P}, \boldsymbol{\omega})$ optimization.

**Remark 1** An interesting observation about the optimal gating parameter $\boldsymbol{\omega}^\star$ is its connection to the correlation matrix $\boldsymbol{R}$, which captures the task correlations in a multitask learning setting. Specifically, the optimal gating given in (14) highlights how $\boldsymbol{\omega}^\star$ depends directly on both the task correlation matrix $\boldsymbol{R}$ and the vector $\boldsymbol{r}$, which encodes the correlations between the tasks and the target task.

**Remark 2** Condition (15) provides a *sufficient* condition for the uniqueness of a fixed point. This implies that whenever $\Delta_{\Sigma} \cdot \Delta_R < M + s_{\min}$, the mapping $h(g(\gamma))$ is a contraction, ensuring the existence of a unique fixed point. However, there may be cases where the mapping $h(g(\gamma))$ does not satisfy Condition (15), yet a unique fixed point (and a unique global minimum) still exists. This is because the Banach Fixed-Point Theorem does not provide a *necessary* condition.

**Corollary 2.** *Suppose* $\Sigma = I$. *Then,* $\Delta_{\Sigma} = 0$, *satisfying Condition* (15), *and we have* $g(\gamma^{\star}) = \frac{1}{d+\sigma^2+1}$, *which yields*

$$P^{\star} = I, \quad \text{and} \quad \omega^{\star} = \left(R + (d + \sigma^2 + 1)I\right)^{-1} r. \tag{17}$$

*Thus, the optimal risk* $\mathcal{L}_{WPGD}^{\star}$ *defined in* (3) *is given by*

$$\mathcal{L}_{WPGD}^{\star} = d + \sigma^2 - d \cdot r^{\top} \left(R + (d + \sigma^2 + 1)I\right)^{-1} r. \tag{18}$$

## 5 Optimization landscape of GLA

In Section 3, we demonstrated that GLA implements a data-dependent WPGD algorithm. Building on this, in Section 4, we analyze the optimization landscape for minimizing the 1-step WPGD risk (c.f. (3)) and show that a unique solution achieves the global minimum of the WPGD algorithm. However, in GLA, the search space for $\omega$ is restricted and data-dependent, meaning that $\mathcal{L}_{WPGD}^{\star}$ in (3) represents the best possible risk a GLA model can achieve. In this section, we analyze the loss landscape for training a 1-layer GLA model and explore the scenarios under which GLA can reach the optimal WPGD risk.

### 5.1 Multi-task prompt model

We consider the following multi-task prompts setting with $K$ correlated tasks $(\beta_k)_{k=1}^K$, and 1 query task $\beta$. For each correlated task, draw a length $n_k$ prompt with IID input-label pairs $\{(x_i^{(k)}, y_i^{(k)})_{i=1}^{n_k}\}_{k=1}^K$ to obtain sequences $(Z_k)_{k=1}^K$ and the query example is given by $z := (x, y \sim \mathcal{N}(x^{\top}\beta, \sigma^2))$. Let $n := \sum_{k=1}^K n_k$. These sequences $(Z_k)_{k=1}^K$ as well as query token $z$ are concatenated to form a single prompt $Z$. Recap the GLA prediction from (1) and let $f_{\text{GLA}}(Z)$ be the GLA prediction as defined in Theorem 1. Additionally, consider the model construction as presented in (8) with $P_q, P_k \in \mathbb{R}^{d \times d}$ being the trainable parameters. Then the GLA optimization problem is described as follows:

$$\mathcal{L}_{\text{GLA}}^{\star} := \min_{P_{k,q}, g} \mathcal{L}_{\text{GLA}}(P_k, P_q, g) \quad \text{where} \quad \mathcal{L}_{\text{GLA}}(P_k, P_q, g) = \mathbb{E}_{\mathcal{D}}\left[(y - f_{\text{GLA}}(Z))^2\right]. \tag{19}$$

Here, $g \in \mathcal{G}$ represents the gating function.

Note that 1) the task vectors $(\beta_k)_{k=1}^K$ are not explicitly shown in the prompt, 2) examples $(x_i^{(k)}, y_i^{(k)})$ are randomly drawn, and 3) the gating function is applied to the tokens/input samples $(Z_k)_{k=1}^K$. Given the above three evidences, the implicit weighting induced by the GLA model varies across different prompts, and it prevents the GLA from learning the optimal weighting.

To address this, we introduce delimiters to mark the boundary of each task. Let $(d_k)_{k=1}^K$ be the delimiters that determine stop of the tasks. Specifically, the final prompt is given by

$$Z = \begin{bmatrix} Z_1^{\top} & d_1 & \cdots & Z_K^{\top} & d_K & z \end{bmatrix}^{\top}. \tag{20}$$

Additionally, to decouple the influence of gating and data, we envision that each token is $z_i = [x_i, y_i, c_i]$ where $c_i \neq 0 \in \mathbb{R}^p$ is the contextual features with $p$ being its dimension and $(x_i, y_i)$ are the data features.

- For task prompts $Z_k$: Contextual features are set to a fixed vector $\bar{d}_0 \neq 0$.

- For delimiters $d_k$: Data features are set to zero (e.g., $x_i = 0$ and $y_i = 0$) so that $d_k = [0_{d+1} \ \bar{d}_k]$ where $\bar{d}_k$ denotes the context vector.

Note that explicit delimiters have been utilized to address real-world problems (Wang et al., 2024; Asai et al., 2022; Dun et al., 2023) due to their ability to improve efficiency and enhance generalization, particularly in task-mixture or multi-document scenarios. To further verify our claim and motivate the introduction of $(d_k)_{k=1}^K$, in Figure 1, we present the results of GLA training with and without delimiters, shown by the red and green curves, respectively. The black dashed curves represent the optimal

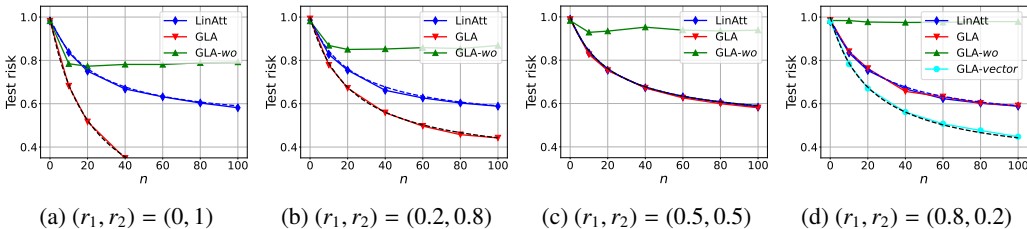

(a) $(r_1, r_2) = (0, 1)$  (b) $(r_1, r_2) = (0.2, 0.8)$  (c) $(r_1, r_2) = (0.5, 0.5)$  (d) $(r_1, r_2) = (0.8, 0.2)$

Figure 1: We consider four different types of model training: `LinAtt` (blue solid): Standard linear attention training. `GLA` (red solid): GLA training using prompts with delimiters (see (20)) and scalar gating. `GLA-wo` (green solid): GLA training using prompts without delimiters and with scalar gating. `GLA-vector` (cyan solid): GLA training using prompts with delimiters and vector gating. The blue and black dashed curves represent the optimal linear attention and WPGD risks from (25) and (18), respectively, as the number of in-context examples $n$ increases. Implementation details are provided in Appendix A.

WPGD loss $\mathcal{L}_{\text{WPGD}}^{\star}$ under different scenarios, and training GLA without delimiters (the green solid curve) performs strictly worse. In contrast, training with delimiters can achieve optimal performance under certain scenarios (see Figures 1a, 1b, and 1c). Theorem 5 in the next section provides a theoretical explanation for these observations, as well as the misalignment seen in Figure 1d. Further discussion and experimental details are provided in Section 5.2 and Appendix A.

## 5.2 Loss landscape of 1-layer GLA

Given the input tokens with extended dimension, to ensure that GLA still implements WPGD as in Theorem 1, we propose the following model construction.

$$\tilde{W}_k = \begin{bmatrix} W_k & 0 \\ 0 & 0 \end{bmatrix}, \quad \tilde{W}_q = \begin{bmatrix} W_q & 0 \\ 0 & 0 \end{bmatrix} \quad \text{and} \quad \tilde{W}_v = \begin{bmatrix} W_v & 0 \\ 0 & 0 \end{bmatrix}. \tag{21}$$

Here, $\tilde{W}_{k,q,v} \in \mathbb{R}^{(d+p+1) \times (d+p+1)}$ and $W_{k,q,v} \in \mathbb{R}^{(d+1) \times (d+1)}$ are constructed via (8). The main idea is to set the last $p$ rows and columns of attention matrices to zeros, ensuring that the delimiters do not affect the final prediction.

**Assumption B.** *Delimiters $\bar{d}_0, \cdots, \bar{d}_K$ are linearly independent, and activation function $\phi(z) : \mathbb{R} \to [0, 1]$ is continuous, satisfying $\phi(-\infty) = 0$ and $\phi(+\infty) = 1$.*

**Assumption C.** *The correlation between context tasks $(\beta_k)_{k=1}^{K}$ and query task $\beta$ satisfies $\mathbb{E}[\beta_i^\top \beta_j] = 0$ and $\mathbb{E}[\beta_i^\top \beta] \leq \mathbb{E}[\beta_j^\top \beta]$ for $1 \leq i \leq j \leq K$.*

Given context examples $\{(X_k, y_k) := (x_i^{(k)}, y_i^{(k)})_{i=1}^{n_k}\}_{k=1}^{K}$, define the concatenated data $(X, y)$ as follows:

$$X = \begin{bmatrix} X_1^\top & \cdots & X_K^\top \end{bmatrix}^\top \in \mathbb{R}^{n \times d} \quad \text{and} \quad y = \begin{bmatrix} y_1^\top & \cdots & y_K^\top \end{bmatrix}^\top \in \mathbb{R}^n. \tag{22}$$

Based on the assumptions above, we are able to establish the equivalence between optimizing 1-layer GLA and optimizing 1-step WPGD predictor under scalar gating.

**Theorem 5** (Scalar Gating). *Recap the loss function $\mathcal{L}_{\text{WPGD}}(P, \omega)$ from (3) with dataset $(X, y)$ defined in (22). Suppose Assumption B holds and consider GLA with scalar gating $g(z) = \phi(w_g^\top z) \mathbf{1} \mathbf{1}^\top$ where $w_g$ is the trainable parameter. Consider input prompt $Z$ defined in (20) and model constructions described in (21). Then the optimal risk $\mathcal{L}_{\text{GLA}}^{\star}$ defined in (19) obeys*

$$\mathcal{L}_{\text{GLA}}^{\star} = \mathcal{L}_{\text{WPGD}}^{\star, \mathcal{W}} \quad \text{where} \quad \mathcal{L}_{\text{WPGD}}^{\star, \mathcal{W}} := \min_{P \in \mathbb{R}^{d \times d}, \omega \in \mathcal{W}} \mathcal{L}_{\text{WPGD}}(P, \omega). \tag{23}$$

*Here, $\mathcal{W} := \left\{ \begin{bmatrix} \omega_1 \mathbf{1}_{n_1}^\top & \cdots & \omega_K \mathbf{1}_{n_K}^\top \end{bmatrix}^\top \in \mathbb{R}^n \mid 0 \leq \omega_i \leq \omega_j \leq 1, \forall 1 \leq i \leq j \leq K \right\}$. Additionally, suppose Assumption C holds and $n_i = n_j$, for any $i, j \in [K]$. Let $\mathcal{L}_{\text{WPGD}}^{\star}$ be the optimal WPGD risk (c.f. (3)). Then $\mathcal{L}_{\text{GLA}}^{\star}$ satisfies*

$$\mathcal{L}_{\text{GLA}}^{\star} = \mathcal{L}_{\text{WPGD}}^{\star}. \tag{24}$$

Assumption B ensures that any $\omega$ in $\mathcal{W}$ can be achieved by an appropriate choice of gating parameters. Furthermore, Assumption C guarantees that the optimal choice of $\omega$ under the WPGD objective lies within the search space $\mathcal{W}$. The proof is provided in Appendix D.1.

In Figure 1, we conduct model training to validate our findings. Consider the setting where $K = 2$ and let $(r_1, r_2) = \left( \mathbb{E}[\beta_1^\top \beta]/d, \mathbb{E}[\beta_2^\top \beta]/d \right)$. In Figures 1a, 1b, and 1c, Assumption C holds, and the

GLA results (shown in solid red) align with the optimal WPGD risk (represented by the dashed black curves), validating (24). However, in Figure 1d, since $r_1 > r_2$, Assumption C does not hold, and as a result, the optimal GLA loss $\mathcal{L}^{\star}_{\text{GLA}}$ obtained from (23) is lower than the optimal WPGD loss $\mathcal{L}^{\star}_{\text{WPGD}}$. Further experimental details are deferred to Appendix A.

**Loss landscape of vector gating.** Till now, much of our discussion has focused on the scalar gating setting. It is important to highlight that, even in the scalar-weighting context, analyzing the WPGD problem remains non-trivial due to the joint optimization over $(\boldsymbol{P}, \omega)$. However, as demonstrated in Theorem 5, scalar gating can only express weightings within the set $\mathcal{W}$. If Assumption C does not hold, $\mathcal{L}^{\star}_{\text{GLA}}$ cannot achieve the optimal WPGD loss (see the misalignment between red solid curve, presenting $\mathcal{L}^{\star}_{\text{GLA}}$, and black dashed curve, presenting $\mathcal{L}^{\star}_{\text{WPGD}}$ in Figure 1d). We argue that vector gating overcomes this limitation by applying distinct weighting mechanisms across different dimensions, facilitating stronger expressivity.

**Theorem 6** (Vector Gating). *Recall input prompt $\boldsymbol{Z}$ from (20) and model constructions from (21) but with $\boldsymbol{W}_v = [\boldsymbol{0}_{(d+1) \times d} \; \boldsymbol{u}]$. Suppose Assumption B holds and consider GLA with vector gating $g(\boldsymbol{z}) = \phi(\boldsymbol{W}_g \boldsymbol{z}) \boldsymbol{1}^{\top}$. Here, $\boldsymbol{u}$ and $\boldsymbol{W}_g$ are trainable parameters. Consider Problem (19), where we employ a vector gating $g(\boldsymbol{z}) = \phi(\boldsymbol{W}_g \boldsymbol{z}) \boldsymbol{1}^{\top}$. Let $\mathcal{L}^{\star}_{\text{GLA-v}}$ denote its optimal risk, and $\mathcal{L}^{\star}_{\text{WPGD}}$ be defined as in (3). Then, the optimal risk obeys $\mathcal{L}^{\star}_{\text{GLA-v}} = \mathcal{L}^{\star}_{\text{WPGD}}$.*

In Theorem 5, the equivalence between $\mathcal{L}^{\star}_{\text{GLA}}$ and $\mathcal{L}^{\star}_{\text{WPGD}}$ is established only when both Assumptions B and C are satisfied. In contrast, Theorem 6 demonstrates that applying vector gating requires only Assumption B to establish $\mathcal{L}^{\star}_{\text{GLA-v}} = \mathcal{L}^{\star}_{\text{WPGD}}$. Specifically, under the bounded activation model of Assumption B, scalar gating is unable to express non-monotonic weighting schemes. For instance, suppose there are two tasks: Even if Task 1 is more relevant to the query, Assumption B will assign a higher weight to examples in Task 2 resulting in sub-optimal prediction. Theorem 6 shows that vector gating can avoid such bottlenecks by potentially encoding tasks in distinct subspaces. To verify these intuitions, in Figure 1d, we train a GLA model with vector gating and results are presented in cyan curve, which outperform the scalar gating results (red solid) and align with the optimal WPGD loss (black dashed).

**Loss landscape of 1-layer linear attention.** Inspired by the fact that linear attention implements all ones gating, that is, $\boldsymbol{G}_i \equiv \boldsymbol{1}$. Consider training a single-layer linear attention and let $f_{\text{ATT}}(\boldsymbol{Z}) := f_{\text{GLA}}(\boldsymbol{Z}, \boldsymbol{G}_i \equiv \boldsymbol{1})$ be its prediction. Let $\mathcal{L}^{\star}_{\text{ATT}}$ be the corresponding optimal risk following (19).

**Corollary 3.** *Consider a single-layer linear attention following model construction in (8) and consider linear data as given in Definition 2. Let $\boldsymbol{R}, \boldsymbol{r}$ be the corresponding correlation matrix and vector as defined in Definition 1. Suppose $\boldsymbol{\Sigma} = \boldsymbol{I}$. Then the optimal risk obeys*

$$\mathcal{L}^{\star}_{\text{ATT}} := \min_{\boldsymbol{P} \in \mathbb{R}^{d \times d}} \mathcal{L}_{\text{WPGD}}(\boldsymbol{P}, \omega = 1) = d + \sigma^2 - \frac{d(\boldsymbol{1}^{\top} \boldsymbol{r})^2}{n(d + \sigma^2 + 1) + \boldsymbol{1}^{\top} \boldsymbol{R} \boldsymbol{1}}. \tag{25}$$

**Corollary 4** (Benefit of Gating). *Consider the same setting as discussed in Corollary 3, and suppose Assumption B holds. Then, we have that $\mathcal{L}^{\star}_{\text{ATT}} \geq \mathcal{L}^{\star}_{\text{GLA}}$. Additionally, if Assumption C holds, we obtain*

$$\mathcal{L}^{\star}_{\text{ATT}} - \mathcal{L}^{\star}_{\text{GLA}} = d \cdot \boldsymbol{r}^{\top} \left( \boldsymbol{R}_+^{-1} - \frac{\boldsymbol{1} \boldsymbol{1}^{\top}}{\boldsymbol{1}^{\top} \boldsymbol{R}_+ \boldsymbol{1}} \right) \boldsymbol{r} \geq 0, \quad \text{where} \quad \boldsymbol{R}_+ := \boldsymbol{R} + \left( d + \sigma^2 + 1 \right) \boldsymbol{I}.$$

The proof of this corollary is directly from (18), (24) and (25). In the Figure 1, blue solid curves represent the linear attention results and blue dashed are the theory curves following (25). The two curves are aligned in all the subfigures, which validate our Corollary 3. More implementation details are deferred to Appendix A.

# 6 Discussion

To summarize, this work offers a fresh theoretical perspective on gated linear attention models through in-context learning by showing that they can emulate data-dependent weighted preconditioned gradient descent (WPGD) algorithms. Our work also reveals how gating is crucial for achieving ICL with stronger data/context adaptivity by demonstrating clear separations between linear attention, scalar-valued gating, and vector-valued gating. We study the optimization landscape of GLA through a connection to the WPGD formulation (3). We have advocated that (3) is a problem of fundamental mathematical interest in its own right, developed the first characterization of its optimization landscape, and showed that it enjoys unique global minima and no other stationary point under mild conditions.

**Limitations and Future Work.** Our analysis is currently limited to characterizing the landscape of scalar gating in GLA models. Extending this framework to vector-valued gating and exploring when delimiters are necessary for learning, as well as investigating the GLA landscape where gates depend on input features, are promising directions for future research.

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

# CONTENTS

# A  IMPLEMENTATION DETAIL

**Data generation.** Consider ICL problem with input in the form of multi-task prompt as described in Section 5.1. In the experiments, we set $K = 2$, dimensions $d = 10$ and $p = 5$, uniform context length $n_1 = n_2 = \bar{n}$, and vary $\bar{n}$ from 0 to 50. Let $(r_1, r_2) := \left( \mathbb{E}[\boldsymbol{\beta}_1^\top \boldsymbol{\beta}]/d, \mathbb{E}[\boldsymbol{\beta}_2^\top \boldsymbol{\beta}]/d \right)$ denote the correlations between in-context tasks $\boldsymbol{\beta}_1, \boldsymbol{\beta}_2$ and query task $\boldsymbol{\beta}$. We generate task vectors as follows:

$$\boldsymbol{\beta}_1, \boldsymbol{\beta}_2 \sim \mathcal{N}(0, \boldsymbol{I}_d) \quad \text{and} \quad \boldsymbol{\beta} \sim \mathcal{N}(r_1 \boldsymbol{\beta}_1 + r_2 \boldsymbol{\beta}_2, (1 - r_1^2 - r_2^2)\boldsymbol{I}_d).$$

Input features are randomly sampled $\boldsymbol{x}_i^{(k)} \sim \mathcal{N}(0, \boldsymbol{I}_d)$ and $y_i^{(k)} = \boldsymbol{\beta}^\top \boldsymbol{x}_i^{(k)}$ ($\sigma = 0$), $k \in \{1, 2\}$. Additionally, delimiters $\bar{\boldsymbol{d}}_0, \cdots, \bar{\boldsymbol{d}}_K$ are randomly sampled from $\mathcal{N}(0, \boldsymbol{I}_p)$.

**Implementation setting.** We train 1-layer linear attention and GLA models for solving multi-prompt ICL problem as described in Section 5.1. For GLA model, we consider sigmoid-type gating function given by scalar gating: $g(z) = \phi(\boldsymbol{w}_g^\top z)\mathbf{1}\mathbf{1}^\top$, or vector gating: $g(z) = \phi(\boldsymbol{W}_g z)\mathbf{1}^\top$ where $\phi(z) = (1 + e^{-z})^{-1}$ is the activation function. Note that although the theoretical results are based on the model constructions (c.f. (8) and (21)), we do not restrict the attention weights in our implementation. We train each model for 10000 iterations with batch size 256 and Adam optimizer with learning rate $10^{-3}$. Similar to the previous work (Li et al., 2024), since our study focuses on the optimization landscape, ICL problems using linear attention/GLA models are non-convex, and experiments are implemented via gradient descent, we repeat 10 model trainings from different model initialization and data sampling (e.g., different choice of delimiters) and results are presented as the minimal test risk among those 10 trails. Results presented have been normalized by $d$.

**Experimental results.** Based on the experimental setting, we can obtain the correlation matrix and vector following Definition 1

$$\boldsymbol{R} = \begin{bmatrix} \mathbf{1}_n \mathbf{1}_n^\top & \mathbf{0} \\ \mathbf{0} & \mathbf{1}_n \mathbf{1}_n^\top \end{bmatrix} \quad \text{and} \quad \boldsymbol{r} = \begin{bmatrix} r_1 \mathbf{1}_n^\top & r_2 \mathbf{1}_n^\top \end{bmatrix}^\top.$$

Then dotted curves display our theoretical results derive using $\boldsymbol{\Sigma} = \boldsymbol{I}$ and $\boldsymbol{R}, \boldsymbol{r}$ above. Specifically, in Figure 1, black dashed curves represent $\mathcal{L}_{\text{WPGD}}^\star$ following (18) and blues dashed curves represent $\mathcal{L}_{\text{GLA}}^\star$ following (25). We consider scenarios where $(r_1, r_2) \in \{(0, 1), (0.2, 0.8), (0.5, 0.5), (0.8, 0.2)\}$ and results are presented in Figures (1a), (1b), (1c) and (1d), respectively.

● `GLA-wo` achieves the worst performance among all the methods. We claim that it is due to the randomness of input tokens as discussed in Section 5.1. Thanks to the introduction of delimiters as described in (20), data and gating is decoupled and a task-dependent weighting is learnt. Hence, `GLA` is able to achieve comparable performance to the optimal one ($\mathcal{L}_{\text{WPGD}}^\star$, red dashed). Note that `GLA-wo`

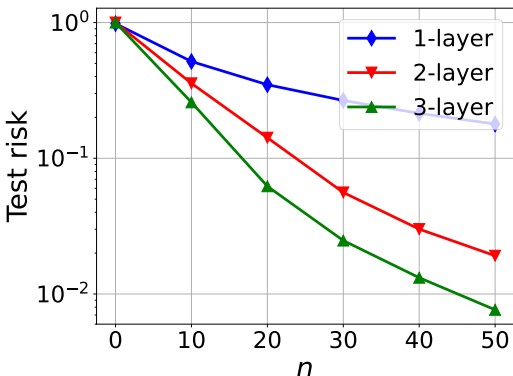

Figure 2: Multi-layer GLA experiments with $(r_1, r_2) = (0, 1)$.

performs even worse than `LinAtt`. It comes from the fact the weighting induced by `GLA-wo` varies over different input prompts and it can not implement all ones weight.

• The alignments between `LinAtt` (blue solid) and blue dashed curves validate our Corollary 3. In Figures 1a, 1b and 1c, the alignments between `GLA` (red solid) and $\mathcal{L}_{\text{WPGD}}$ (black dashed) verify our Theorem 5, specifically, Equation 24. While in 1c and 1d, `GLA` achieves the same performance as `LinAtt`. It is due to the fact that `GLA` can not weight the history higher than its present. Then the equal-weighting, e.g., $\omega = 1$, is the optimal weighting given such constraint. What's more, the alignment between `GLA-vector` (cyan curves) and red dashed in Figure 1d validates our vector gating theorem in Theorem 6.

### A.1 MULTI-LAYER EXPERIMENTS

In this section, we present additional experiments on multi-layer GLA models. We adopt the same experimental setup as described in Figure 1a and Appendix A, with parameters set to $(r_1, r_2) = (0, 1)$. The results are displayed in Figure 2, where the blue, red, and green curves correspond to the performance of one-, two-, and three-layer GLA models, respectively, with the $y$-axis presented in log-scale. According to Theorem 2, an $L$-layer GLA performs $L$ steps of WPGD, suggesting that deeper models should yield improved predictive performance. The experimental findings in Figure 2 align with the theoretical predictions of Theorem 2.

## B GLA ⇔ WPGD

### B.1 PROOF OF THEROEM 1

Recap the problem settings from Section 2 where in-context samples are given by

$$Z = [z_1 \cdots z_n \, z_{n+1}]^\top = \begin{bmatrix} x_1 & \cdots & x_n & x_{n+1} \\ y_1 & \cdots & y_n & 0 \end{bmatrix}^\top$$

and let the value, key and query embeddings at time $i$ be

$$v_i = W_v z_i, \quad k_i = W_k z_i, \quad \text{and} \quad q_i = W_q z_i.$$

Then we can rewrite the GLA output (c.f. (1)) as follows:

$$o_i = S_i q_i \quad \text{and} \quad S_i = G_i \odot S_{i-1} + v_i k_i^\top$$

$$= \sum_{j=1}^{i} G_{j:i} \odot v_j k_j^\top$$

where we define

$$G_{j:i} = G_{j+1} \odot G_{j+2} \cdots G_i, \quad j < i, \quad \text{and} \quad G_{i:i} = \mathbf{1}\mathbf{1}^\top.$$

Consider the prediction based on the last token, then we obtain

$$o_{n+1} = S_{n+1} q_{n+1} \quad \text{and} \quad S_{n+1} = \sum_{j=1}^{n+1} G_{j:n+1} \odot v_j k_j^\top.$$

**Construction 1:** Recall the model construction from (8) where

$$W_k = \begin{bmatrix} P_k^\top & \mathbf{0} \\ \mathbf{0}^\top & 0 \end{bmatrix}, \quad W_q = \begin{bmatrix} P_q^\top & \mathbf{0} \\ \mathbf{0}^\top & 0 \end{bmatrix} \quad \text{and} \quad W_v = \begin{bmatrix} \mathbf{0}_{d\times d} & \mathbf{0} \\ \mathbf{0}^\top & 1 \end{bmatrix}. \tag{26}$$

Then, given each token $z_i = [x_i^\top \; y_i]^\top$, $i \in [n]$, single-layer GLA returns

$$v_i = \begin{bmatrix} \mathbf{0} \\ y_i \end{bmatrix}, \quad k_i = \begin{bmatrix} P_k^\top x_i \\ 0 \end{bmatrix}, \quad \text{and} \quad q_i = \begin{bmatrix} P_q^\top x_i \\ 0 \end{bmatrix},$$

and we obtain

$$v_i k_i^\top = \begin{bmatrix} \mathbf{0}_{d\times d} & \mathbf{0} \\ y_i x_i^\top P_k & 0 \end{bmatrix}, \quad i \le n, \quad \text{and} \quad v_{n+1} k_{n+1}^\top = \mathbf{0}_{(d+1)\times(d+1)}.$$

Therefore, since only $d$ entries in $v_i k_i^\top$ matrix are nonzero, given $\odot$ as the Hadamard product, only the corresponding $d$ entries in all $G_i$ matrices are useful. Based on this observation, let

$$G_i = \begin{bmatrix} * & * \\ g_i^\top & * \end{bmatrix} \quad \text{and} \quad G_{j:i} = \begin{bmatrix} * & * \\ g_{j:i}^\top & * \end{bmatrix}$$

where $g_{j:i} = g_{j+1} \odot g_{j+2} \cdots g_i \in \mathbb{R}^d$ for $j < i$ and $g_{i:i} = \mathbf{1}_d$.

Combing all together, and letting $X = [x_1 \; x_2 \; \cdots \; x_n]^\top$ and $y = [y_1 \; y_2 \; \cdots \; y_n]^\top$, we obtain

$$o_{n+1} = S_{n+1} q_{n+1} = \begin{bmatrix} \mathbf{0}_{d\times d} & \mathbf{0} \\ \sum_{j=1}^n y_j x_j^\top P_k \odot g_{j:n+1}^\top & 0 \end{bmatrix} \begin{bmatrix} P_q^\top x \\ 0 \end{bmatrix} = \begin{bmatrix} \mathbf{0} \\ x^\top P_q (XP_k \odot \Omega)^\top y \end{bmatrix}$$

where

$$\Omega = \begin{bmatrix} g_{1:n+1} & g_{2:n+1} & \cdots & g_{n:n+1} \end{bmatrix} \in \mathbb{R}^{n\times d}.$$

Then if taking the last entry of $o_{n+1}$ as final prediction, we get

$$\hat{y} := o_{n+1,d+1} = x^\top \hat{\beta} \quad \text{where} \quad \hat{\beta} = P_q (XP_k \odot \Omega)^\top y.$$

It completes the proof of Theorem 1.

**Construction 2:** Based on the construction given in (26), only $d$ elements of $G_i$ matrices are useful. One might ask about the effect of other entries of $G_i$. Therefore, in the following, we introduce an other model construction showing that different row of $G_i$ implements WPGD with different weighting. Similarly, let $W_k, W_q$ be the same as (26) but with $W_v$ constructed by

$$W_v = \begin{bmatrix} \mathbf{0}_{(d+1)\times d} & u \end{bmatrix} \quad \text{where} \quad u = [u_1 \; u_2 \; \cdots \; u_{d+1}]^\top \in \mathbb{R}^{d+1}.$$

Then the value embeddings have the form of $v_i = y_i u$, which gives

$$v_i k_i^\top = u \begin{bmatrix} y_i x_i^\top P_k & 0 \end{bmatrix}.$$

Next, let

$$G_i = \begin{bmatrix} (g_i^1)^\top & * \\ (g_i^2)^\top & * \\ \vdots & \vdots \\ (g_i^{d+1})^\top & * \end{bmatrix} \quad \text{and} \quad G_{j:i} = \begin{bmatrix} (g_{j:i}^1)^\top & * \\ (g_{j:i}^2)^\top & * \\ \vdots & \vdots \\ (g_{j:i}^{d+1})^\top & * \end{bmatrix}$$

where $g_i^{i'} \in \mathbb{R}^d$ corresponds to the $i'$-th row of $G_i$ and $g_{j:i}^{i'} = g_{j+1}^{i'} \odot g_{j+1}^{i'} \cdots g_i^{i'}$. Then we get the output

$$o_{n+1} = \begin{bmatrix} \sum_{j=1}^n u_1 y_j x_j^\top P_k \odot (g_{j:n+1}^1)^\top & 0 \\ \sum_{j=1}^n u_2 y_j x_j^\top P_k \odot (g_{j:n+1}^2)^\top & 0 \\ \vdots & \\ \sum_{j=1}^n u_{d+1} y_j x_j^\top P_k \odot (g_{j:n+1}^{d+1})^\top & 0 \end{bmatrix} \begin{bmatrix} P_q^\top x \\ 0 \end{bmatrix} = \begin{bmatrix} x^\top P_q (XP_k \odot \Omega_1)^\top y \\ x^\top P_q (XP_k \odot \Omega_2)^\top y \\ \vdots \\ x^\top P_q (XP_k \odot \Omega_{d+1})^\top y \end{bmatrix}$$

where

$$\Omega_i = u_i \begin{bmatrix} g_{1:n+1}^i & g_{2:n+1}^i & \cdots & g_{n:n+1}^i \end{bmatrix} \in \mathbb{R}^{n\times d}, \quad i \le d+1.$$

Therefore, consider $(d+1)$-dimensional output $o_{n+1}$. Each entry implements a 1-step WPGD with same preconditioners $P_k, P_q$ and different weighting matrices $\Omega$'s. The weighting matrix of $i$'th entry is determined by the $i$'th row of all gating matrices. Note that if consider the last entry of $o_{n+1}$ as prediction, it returns the same result as Construction 1 above, where only last rows of $G_i$'s are useful.

Additionally, suppose that the final prediction $\hat{y}$ is given after a linear head $\boldsymbol{h}$, that is, $\hat{y} = \boldsymbol{h}^\top \boldsymbol{o}_{n+1}$, and let $\boldsymbol{h} = [h_1 \ h_2 \ \cdots \ h_{d+1}]^\top \in \mathbb{R}^{d+1}$. Then

$$\hat{y} = \boldsymbol{h}^\top \boldsymbol{o}_{n+1} = \boldsymbol{x}^\top \boldsymbol{P}_q \left( \boldsymbol{X}\boldsymbol{P}_k \odot \bar{\boldsymbol{\Omega}} \right)^\top \boldsymbol{y} \tag{27}$$

where

$$\bar{\boldsymbol{\Omega}} = \sum_{i=1}^{d+1} h_i \boldsymbol{\Omega}_i = \sum_{i=1}^{d+1} h_i u_i \left[ \boldsymbol{g}^i_{1:n+1} \quad \boldsymbol{g}^i_{2:n+1} \quad \cdots \quad \boldsymbol{g}^i_{n:n+1} \right] \in \mathbb{R}^{n \times d}. \tag{28}$$

Then, single-layer GLA still returns 1-step WPGD with updated weighting matrix.

## B.2 Proof of Theorem 2

**Theorem 7** (Extended version of Theorem 2). *Consider an $L$-layer GLA with $\ell$'th layer parameterized by $\boldsymbol{P}_{k,\ell}, \boldsymbol{P}_{q,\ell} \in \mathbb{R}^{d \times d}$ as in (8) and with corresponding gating vectors $\boldsymbol{g}^\ell_i$, $i \in [n+1], \ell \in [L]$. Let $\hat{y}_{i,\ell}$ be the $(d+1)$'th entry of the $i$'th token of the $\ell$'th layer input (or $(\ell-1)$'th layer output after residual). Additionally, denote $\boldsymbol{\Omega}_\ell = [\boldsymbol{g}^\ell_{1:n+1} \ \cdots \ \boldsymbol{g}^\ell_{n:n+1}]^\top$ and $\bar{\boldsymbol{X}}_\ell = \boldsymbol{X}\boldsymbol{P}_{k,\ell} \odot \boldsymbol{\Omega}_\ell$. Let $\boldsymbol{B}_\ell = [\boldsymbol{\beta}_{1,\ell} \ \cdots \ \boldsymbol{\beta}_{n,\ell}]^\top$ where $\boldsymbol{\beta}_{i,0} = \boldsymbol{0}$ for $i \in [n+1]$ and $\boldsymbol{M}_i = \begin{bmatrix} \boldsymbol{I}_i & \boldsymbol{0} \\ \boldsymbol{0} & \boldsymbol{0} \end{bmatrix} \in \mathbb{R}^{n \times n}$. Then it satisfies that for*

- *$i \le n$, $\hat{y}_{i,\ell} = y_i - \boldsymbol{x}_i^\top \boldsymbol{\beta}_{i,\ell-1}$ where $\boldsymbol{\beta}_{i,\ell} = \boldsymbol{\beta}_{i,\ell-1} + \boldsymbol{P}_{q,\ell} \left( \nabla_{i,\ell} \oslash \boldsymbol{g}^\ell_{i:n+1} \right)$*

- *and $\hat{y}_{n+1,\ell} = \boldsymbol{x}^\top \boldsymbol{\beta}_{\ell-1}$ where $\boldsymbol{\beta}_\ell = (1 + \alpha_\ell)\boldsymbol{\beta}_{\ell-1} + \boldsymbol{P}_{q,\ell} \left( \nabla_{n,\ell} \oslash \boldsymbol{g}^\ell_{n+1} \right)$ and $\alpha_\ell = \boldsymbol{x}^\top \boldsymbol{P}_{q,\ell} \boldsymbol{P}_{k,\ell}^\top \boldsymbol{x}$.*

*Here, we define $\nabla_{i,\ell} = \bar{\boldsymbol{X}}_\ell^\top \boldsymbol{M}_i ((\boldsymbol{X} \odot \boldsymbol{B}_{\ell-1})\boldsymbol{1} - \boldsymbol{y})$.*

*Proof.* Recapping the model construction from (8) and following the same analysis in Appendix B.1, for $i \le n$, we obtain

$$\boldsymbol{S}_i = \begin{bmatrix} \boldsymbol{0}_{d \times d} & \boldsymbol{0} \\ \sum_{j=1}^i y_j \boldsymbol{x}_j^\top \boldsymbol{P}_k \odot \boldsymbol{g}_{j:i}^\top & 0 \end{bmatrix}.$$

Additionally, recap that we have

$$\boldsymbol{M}_i = \begin{bmatrix} \boldsymbol{I}_i & \boldsymbol{0} \\ \boldsymbol{0} & \boldsymbol{0} \end{bmatrix} \quad \text{and} \quad \boldsymbol{\Omega} = \begin{bmatrix} \boldsymbol{g}_{i:n+1} & \cdots & \boldsymbol{g}_{n:n+1} \end{bmatrix}.$$

Let $\oslash$ denote Hadamard division. Then

$$\sum_{j=1}^i y_j \boldsymbol{P}_k^\top \boldsymbol{x}_j \odot \boldsymbol{g}_{j:i} = \left( \sum_{j=1}^i y_j \boldsymbol{P}_k^\top \boldsymbol{x}_j \odot \boldsymbol{g}_{j:n+1} \right) \oslash \boldsymbol{g}_{i:n+1}$$

$$= (\boldsymbol{X}\boldsymbol{P}_k \odot \boldsymbol{\Omega})^\top \boldsymbol{M}_i \boldsymbol{y} \oslash \boldsymbol{g}_{i:n+1},$$

Therefore,

$$\boldsymbol{o}_i = \boldsymbol{S}_i \boldsymbol{q}_i = \begin{bmatrix} \boldsymbol{0}_{d \times d} & \boldsymbol{0} \\ ((\boldsymbol{X}\boldsymbol{P}_k \odot \boldsymbol{\Omega})^\top \boldsymbol{M}_i \boldsymbol{y} \oslash \boldsymbol{g}_{i:n+1})^\top & 0 \end{bmatrix} \begin{bmatrix} \boldsymbol{P}_q^\top \boldsymbol{x}_i \\ 0 \end{bmatrix} = \begin{bmatrix} \boldsymbol{0} \\ \boldsymbol{x}_i^\top \boldsymbol{P}_q \left( \bar{\boldsymbol{X}}^\top \boldsymbol{M}_i \boldsymbol{y} \oslash \boldsymbol{g}_{i:n+1} \right) \end{bmatrix}. \tag{29}$$

where we define $\bar{\boldsymbol{X}} := \boldsymbol{X}\boldsymbol{P}_k \odot \boldsymbol{\Omega}$. Similarly, we can get the last token output

$$\boldsymbol{o}_{n+1} = \begin{bmatrix} \boldsymbol{0} \\ \boldsymbol{x}^\top \boldsymbol{P}_q \left( \bar{\boldsymbol{X}}^\top \boldsymbol{y} \oslash \boldsymbol{g}_{n+1} \right) \end{bmatrix}. \tag{30}$$

Next, we consider the multi-layer GLA model. To begin with, let us define the input and output of $\ell$'th layer as

$$\boldsymbol{Z}_\ell = \begin{bmatrix} \boldsymbol{z}_{1,\ell} & \cdots & \boldsymbol{z}_{n,\ell} & \boldsymbol{z}_{n+1,\ell} \end{bmatrix}^\top \in \mathbb{R}^{(n+1) \times (d+1)},$$

$$\boldsymbol{O}_\ell = \begin{bmatrix} \boldsymbol{o}_{1,\ell} & \cdots & \boldsymbol{o}_{n,\ell} & \boldsymbol{o}_{n+1,\ell} \end{bmatrix}^\top \in \mathbb{R}^{(n+1) \times (d+1)},$$

where $\boldsymbol{Z}_1 = \boldsymbol{Z}$. Then, given the residual connection of each layer, the input of $(\ell+1)$'th layer is given by

$$\boldsymbol{Z}_{\ell+1} = \boldsymbol{Z}_\ell + \boldsymbol{O}_\ell. \tag{31}$$

Note that $\mathbf{Z}_{\ell+1}$ is also the output of $\ell$'th layer after residual. Recall (29) which implies that the first $d$ dimension of the output $\mathbf{o}_i$ for all tokens $i \in [n+1]$ is zero. Therefore, the first $d$ dimension of $z_{i,\ell}$ keeps the same as $\mathbf{x}_i$ and let us write the input of $\ell$'th layer (also the output of $(\ell-1)$'th layer after residual) as

$$\mathbf{Z}_\ell = \begin{bmatrix} \mathbf{x}_1 & \cdots & \mathbf{x}_n & \mathbf{x} \\ \hat{y}_{1,\ell} & \cdots & \hat{y}_{n,\ell} & \hat{y}_{n+1,\ell} \end{bmatrix}^\top, \tag{32}$$

and $\hat{y}_{i,1} = y_i$ for $i \in [n]$ and $\hat{y}_{n+1,1} = 0$. Suppose that the $\ell$'th layer is parameterized by $(\mathbf{P}_{q,\ell}, \mathbf{P}_{k,\ell})$ and let $\mathbf{Z}_\ell$ be its input. Additionally, suppose the gating matrices for $\ell$'th layer, $i$'th token is

$$\mathbf{G}_i^\ell = \begin{bmatrix} * & * \\ (\mathbf{g}_i^\ell)^\top & * \end{bmatrix}.$$

- We first study $\hat{y}_{i,\ell}$ for $i \le n$. Following (29), we obtain the output at time $i$

$$\mathbf{o}_{i,\ell} = \begin{bmatrix} \mathbf{0} \\ \mathbf{x}_i^\top \mathbf{P}_{q,\ell} \left( \bar{\mathbf{X}}_\ell^\top \mathbf{M}_i \hat{\mathbf{y}}_\ell \oslash \mathbf{g}_{i:n+1}^\ell \right) \end{bmatrix}$$

where $\bar{\mathbf{X}}_\ell := \mathbf{X}\mathbf{P}_{k,\ell} \odot \mathbf{\Omega}_\ell$ and

$$\hat{\mathbf{y}}_\ell = [\hat{y}_{1,\ell} \quad \cdots \quad \hat{y}_{n,\ell}]^\top \in \mathbb{R}^n,$$

$$\mathbf{\Omega}_\ell = \begin{bmatrix} \mathbf{g}_{1:n+1}^\ell & \mathbf{g}_{2:n+1}^\ell & \cdots & \mathbf{g}_{n:n+1}^\ell \end{bmatrix}^\top \in \mathbb{R}^{n \times d}.$$

Following the residual connection as in (31), we have $z_{i,\ell+1} = z_{i,\ell} + \mathbf{o}_{i,\ell}$ and hence

$$\hat{y}_{i,\ell+1} = \hat{y}_{i,\ell} + \mathbf{x}_i^\top \mathbf{P}_{q,\ell} \left( \bar{\mathbf{X}}_\ell^\top \mathbf{M}_i \hat{\mathbf{y}}_\ell \oslash \mathbf{g}_{i:n+1}^\ell \right). \tag{33}$$

Now consider the algorithm given in the theorem statement where $\hat{y}_{i,\ell} = y_i - \mathbf{x}_i^\top \boldsymbol{\beta}_{i,\ell-1}$ and $\boldsymbol{\beta}_{i,\ell} = \boldsymbol{\beta}_{i,\ell-1} + \mathbf{P}_{q,\ell} \left( \bar{\mathbf{X}}_\ell^\top \mathbf{M}_i((\mathbf{X} \odot \mathbf{B}_{\ell-1})\mathbf{1} - \mathbf{y}) \oslash \mathbf{g}_{i:n+1}^\ell \right)$, which gives

$$\hat{y}_{i,\ell+1} = y_i - \mathbf{x}_i^\top \boldsymbol{\beta}_{i,\ell}$$
$$\hat{y}_{i,\ell} = y_i - \mathbf{x}_i^\top \boldsymbol{\beta}_{i,\ell-1}. \tag{34}$$

Then

$$\hat{y}_{i,\ell+1} - \hat{y}_{i,\ell} = -\mathbf{x}_i^\top (\boldsymbol{\beta}_{i,\ell} - \boldsymbol{\beta}_{i,\ell-1})$$
$$= -\mathbf{x}_i^\top \mathbf{P}_{q,\ell} \left( \bar{\mathbf{X}}_\ell^\top \mathbf{M}_i((\mathbf{X} \odot \mathbf{B}_{\ell-1})\mathbf{1} - \mathbf{y}) \oslash \mathbf{g}_{i:n+1}^\ell \right)$$
$$= \mathbf{x}_i^\top \mathbf{P}_{q,\ell} \left( \bar{\mathbf{X}}_\ell^\top \mathbf{M}_i \hat{\mathbf{y}}_\ell \oslash \mathbf{g}_{i:n+1}^\ell \right). \tag{35}$$

The last equation uses (34), that

$$(\mathbf{X} \odot \mathbf{B}_{\ell-1})\mathbf{1} = [\mathbf{x}_1^\top \boldsymbol{\beta}_{1,\ell} \quad \cdots \quad \mathbf{x}_n^\top \boldsymbol{\beta}_{n,\ell}]^\top \implies (\mathbf{X} \odot \mathbf{B}_{\ell-1})\mathbf{1} - \mathbf{y} = -\hat{\mathbf{y}}_\ell. \tag{36}$$

The equality between (33) and (35) completes the proof for $i \in [n]$.

- Next, we consider the last token output, that is $i = n+1$. In the following, we remove the subscript $n+1$ from some notations for simplification.

Similarly, we get the $(n+1)$'th output of $\ell$'th layer

$$\mathbf{o}_{n+1,\ell} = \begin{bmatrix} \mathbf{0} \\ \mathbf{x}^\top \mathbf{P}_{q,\ell} \left( \bar{\mathbf{X}}_\ell^\top \hat{\mathbf{y}}_\ell \oslash \mathbf{g}_{n+1}^\ell \right) \end{bmatrix} + \begin{bmatrix} \mathbf{0} \\ \mathbf{x}^\top \mathbf{P}_{q,\ell} \mathbf{P}_{k,\ell}^\top \mathbf{x} \cdot \hat{y}_{n+1,\ell} \end{bmatrix}$$

where the second term comes from the fact that $\hat{y}_{n+1,\ell} \ne 0$ for $\ell \ne 0$.

Let $\alpha_\ell := \mathbf{x}^\top \mathbf{P}_{q,\ell} \mathbf{P}_{k,\ell}^\top \mathbf{x}$. Given $\mathbf{Z}_{\ell+1} = \mathbf{Z}_\ell + \mathbf{O}_\ell$, we obtain

$$\hat{y}_{n+1,\ell+1} = \hat{y}_{n+1,\ell} + \mathbf{x}^\top \mathbf{P}_{q,\ell} \left( \bar{\mathbf{X}}_\ell^\top \hat{\mathbf{y}}_\ell \oslash \mathbf{g}_{n+1}^\ell \right) + \alpha_\ell \cdot \hat{y}_{n+1,\ell}$$
$$= (1 + \alpha_\ell)\hat{y}_{n+1,\ell} + \mathbf{x}^\top \mathbf{P}_{q,\ell} \left( \bar{\mathbf{X}}_\ell^\top \hat{\mathbf{y}}_\ell \oslash \mathbf{g}_{n+1}^\ell \right). \tag{37}$$

Now, consider the algorithm given in the theorem statement where $\hat{y}_{n+1,\ell} = -\mathbf{x}^\top \boldsymbol{\beta}_{n+1,\ell-1}$ and $\boldsymbol{\beta}_{n+1,\ell} = (1 + \alpha_\ell)\boldsymbol{\beta}_{n+1,\ell-1} + \mathbf{P}_{q,\ell} \left( \bar{\mathbf{X}}_\ell^\top ((\mathbf{X} \odot \mathbf{B}_{\ell-1})\mathbf{1} - \mathbf{y}) \oslash \mathbf{g}_{n+1}^\ell \right)$, which gives

$$\hat{y}_{n+1,\ell+1} = -\mathbf{x}^\top \boldsymbol{\beta}_{n+1,\ell}$$
$$\hat{y}_{n+1,\ell} = -\mathbf{x}^\top \boldsymbol{\beta}_{n+1,\ell-1}.$$

Then

$$\hat{y}_{n+1,\ell+1} - (1 + \alpha_\ell)\hat{y}_{n+1,\ell} = -\boldsymbol{x}^\top \left(\boldsymbol{\beta}_{n+1,\ell} - (1 + \alpha_\ell)\boldsymbol{\beta}_{n+1,\ell-1}\right)$$

$$= -\boldsymbol{x}^\top \boldsymbol{P}_{q,\ell} \left(\bar{\boldsymbol{X}}_\ell^\top ((\boldsymbol{X} \odot \boldsymbol{B}_{\ell-1})\mathbf{1} - \boldsymbol{y}) \oslash \boldsymbol{g}_{n+1}^\ell\right)$$

$$= \boldsymbol{x}^\top \boldsymbol{P}_{q,\ell} \left(\bar{\boldsymbol{X}}_\ell^\top \hat{\boldsymbol{y}}_\ell \oslash \boldsymbol{g}_{n+1}^\ell\right)$$

which is the same as (37) by using the fact from (36). $\qquad\square$

## C  OPTIMIZATION LANDSCAPE OF WPGD

### C.1  PROOF OF THEOREM 3

*Proof.* Recapping the objective from (3) and following Definition 2, we have

$$\mathcal{L}(\boldsymbol{P}, \boldsymbol{\omega}) = \mathbb{E}\left[\left(y - \boldsymbol{x}^\top \boldsymbol{P} \boldsymbol{X}(\boldsymbol{\omega} \odot \boldsymbol{y})\right)^2\right]$$

$$= \mathbb{E}\left[y^2\right] - 2\mathbb{E}\left[y\boldsymbol{x}^\top \boldsymbol{P} \boldsymbol{X}(\boldsymbol{\omega} \odot \boldsymbol{y})\right] + \mathbb{E}\left[\left(\boldsymbol{x}^\top \boldsymbol{P} \boldsymbol{X}(\boldsymbol{\omega} \odot \boldsymbol{y})\right)^2\right].$$

Let $y = \boldsymbol{x}^\top \boldsymbol{\beta} + \xi$ and $y_i = \boldsymbol{x}_i^\top \boldsymbol{\beta}_i + \xi_i$, for $i \in [n]$, where $\xi, \xi_i \sim \mathcal{N}(0, \sigma^2)$ are i.i.d. Then,

$$\mathbb{E}[y^2] = \mathbb{E}[(\boldsymbol{x}^\top \boldsymbol{\beta} + \xi)^2] = \mathtt{tr}\,(\boldsymbol{\Sigma}) + \sigma^2,$$

and

$$\mathbb{E}\left[y\boldsymbol{x}^\top \boldsymbol{P} \boldsymbol{X}(\boldsymbol{\omega} \odot \boldsymbol{y})\right] = \mathbb{E}\left[(\boldsymbol{\beta}^\top \boldsymbol{x} + \xi)\boldsymbol{x}^\top \boldsymbol{P} \sum_{i=1}^n \omega_i \boldsymbol{x}_i(\boldsymbol{x}_i^\top \boldsymbol{\beta}_i + \xi_i)\right]$$

$$= \mathbb{E}\left[\boldsymbol{\beta}^\top \boldsymbol{x} \boldsymbol{x}^\top \boldsymbol{P} \sum_{i=1}^n \omega_i \boldsymbol{x}_i \boldsymbol{x}_i^\top \boldsymbol{\beta}_i\right]$$

$$= \mathtt{tr}\left(\boldsymbol{\Sigma} \boldsymbol{P} \boldsymbol{\Sigma} \sum_{i=1}^n \omega_i \mathbb{E}\left[\boldsymbol{\beta}_i \boldsymbol{\beta}^\top\right]\right)$$

$$= \mathtt{tr}\left(\boldsymbol{\Sigma}^2 \boldsymbol{P}\right) \boldsymbol{\omega}^\top \boldsymbol{r}.$$

Here, the last equality comes from the fact that since $\boldsymbol{\beta}_i - r_{ij}\boldsymbol{\beta}_j$ is independent of $\boldsymbol{\beta}_j$ for $i, j \in [n+1]$ following Definition 1, we have $\mathbb{E}\left[\boldsymbol{\beta}_i \boldsymbol{\beta}^\top\right] = r_{i,n+1}\boldsymbol{I}_d$ and $\sum_{i=1}^n \omega_i \mathbb{E}\left[\boldsymbol{\beta}_i \boldsymbol{\beta}^\top\right]$ returns $\boldsymbol{\omega}^\top \boldsymbol{r} \cdot \boldsymbol{I}_d$.

Hence,

$$\mathbb{E}\left[\left(\boldsymbol{x}^\top \boldsymbol{P} \boldsymbol{X}(\boldsymbol{\omega} \odot \boldsymbol{y})\right)^2\right] = \mathbb{E}\left[\boldsymbol{x}^\top \boldsymbol{P}\left(\sum_{i=1}^n \omega_i(\boldsymbol{x}_i^\top \boldsymbol{\beta}_i + \xi_i)\boldsymbol{x}_i\right)\left(\sum_{i=1}^n \omega_i \boldsymbol{x}_i^\top(\boldsymbol{x}_i^\top \boldsymbol{\beta}_i + \xi_i)\right)\boldsymbol{P}^\top \boldsymbol{x}\right]$$

$$= \mathtt{tr}\left(\boldsymbol{P}^\top \boldsymbol{\Sigma} \boldsymbol{P} \mathbb{E}\left[\sum_{i=1}^n \omega_i^2(\boldsymbol{x}_i^\top \boldsymbol{\beta}_i + \xi_i)^2 \boldsymbol{x}_i \boldsymbol{x}_i^\top\right]\right)$$

$$+ \mathtt{tr}\left(\boldsymbol{P}^\top \boldsymbol{\Sigma} \boldsymbol{P} \mathbb{E}\left[\sum_{i \neq j} \omega_i \omega_j(\boldsymbol{x}_i^\top \boldsymbol{\beta}_i + \xi_i)\boldsymbol{x}_i \boldsymbol{x}_j^\top(\boldsymbol{x}_j^\top \boldsymbol{\beta}_j + \xi_j)\right]\right),$$

where

$$\mathtt{tr}\left(\boldsymbol{P}^\top \boldsymbol{\Sigma} \boldsymbol{P} \mathbb{E}\left[\sum_{i=1}^n \omega_i^2(\boldsymbol{x}_i^\top \boldsymbol{\beta}_i + \xi_i)^2 \boldsymbol{x}_i \boldsymbol{x}_i^\top\right]\right) = \mathtt{tr}\left(\boldsymbol{P}^\top \boldsymbol{\Sigma} \boldsymbol{P} \mathbb{E}\left[\sum_{i=1}^n \omega_i^2(\boldsymbol{x}_i^\top \boldsymbol{\beta}_i \boldsymbol{\beta}_i^\top \boldsymbol{x}_i + \sigma^2)\boldsymbol{x}_i \boldsymbol{x}_i^\top\right]\right)$$

$$= \|\boldsymbol{\omega}\|_{\ell_2}^2 \, \mathtt{tr}\left(\boldsymbol{P}^\top \boldsymbol{\Sigma} \boldsymbol{P}\left(\mathbb{E}\left[\boldsymbol{x} \boldsymbol{x}^\top \boldsymbol{x} \boldsymbol{x}^\top\right] + \sigma^2 \boldsymbol{\Sigma}\right)\right)$$

$$= \|\boldsymbol{\omega}\|_{\ell_2}^2 \left(\mathtt{tr}\left(\boldsymbol{\Sigma} \boldsymbol{P}^\top \boldsymbol{\Sigma} \boldsymbol{P}\right)\left(\mathtt{tr}\,(\boldsymbol{\Sigma}) + \sigma^2\right) + \mathtt{tr}\left(\boldsymbol{\Sigma}^2 \boldsymbol{P}^\top \boldsymbol{\Sigma} \boldsymbol{P}\right)\right),$$

and

$$\mathtt{tr}\left(\boldsymbol{P}^\top \boldsymbol{\Sigma} \boldsymbol{P} \mathbb{E}\left[\sum_{i \neq j} \omega_i \omega_j(\boldsymbol{x}_i^\top \boldsymbol{\beta}_i + \xi_i)\boldsymbol{x}_i \boldsymbol{x}_j^\top(\boldsymbol{x}_j^\top \boldsymbol{\beta}_j + \xi_j)\right]\right) = \mathtt{tr}\left(\boldsymbol{P}^\top \boldsymbol{\Sigma} \boldsymbol{P} \mathbb{E}\left[\sum_{i \neq j} \omega_i \omega_j \boldsymbol{x}_i \boldsymbol{x}_i^\top \boldsymbol{\beta}_i \boldsymbol{\beta}_j^\top \boldsymbol{x}_j \boldsymbol{x}_j^\top\right]\right)$$

$$= \mathtt{tr}\left(\boldsymbol{\Sigma}^2 \boldsymbol{P}^\top \boldsymbol{\Sigma} \boldsymbol{P}\right) \boldsymbol{\omega}^\top \boldsymbol{R} \boldsymbol{\omega}.$$

Combining all together and letting $M := \text{tr}(\mathbf{\Sigma}) + \sigma^2$, we obtain

$$
\begin{aligned}
\mathcal{L}(\mathbf{P}, \boldsymbol{\omega}) = M &- 2\text{tr}\left(\mathbf{\Sigma}^2 \mathbf{P}\right) \boldsymbol{\omega}^\top \mathbf{r} \\
&+ M \|\boldsymbol{\omega}\|_{\ell_2}^2 \, \text{tr}\left(\mathbf{\Sigma} \mathbf{P}^\top \mathbf{\Sigma} \mathbf{P}\right) + (\|\boldsymbol{\omega}\|_{\ell_2}^2 + \boldsymbol{\omega}^\top \mathbf{R} \boldsymbol{\omega})\text{tr}\left(\mathbf{\Sigma}^2 \mathbf{P}^\top \mathbf{\Sigma} \mathbf{P}\right).
\end{aligned}
\tag{38}
$$

For simplicity, and without loss of generality, let

$$
\tilde{\mathbf{P}} = \sqrt{\mathbf{\Sigma}} \mathbf{P} \sqrt{\mathbf{\Sigma}}.
\tag{39}
$$

Then, we obtain

$$
\begin{aligned}
\mathcal{L}(\tilde{\mathbf{P}}, \boldsymbol{\omega}) = M &- 2\text{tr}\left(\mathbf{\Sigma} \tilde{\mathbf{P}}\right) \boldsymbol{\omega}^\top \mathbf{r} \\
&+ M \|\boldsymbol{\omega}\|_{\ell_2}^2 \, \text{tr}\left(\tilde{\mathbf{P}}^\top \tilde{\mathbf{P}}\right) + (\|\boldsymbol{\omega}\|_{\ell_2}^2 + \boldsymbol{\omega}^\top \mathbf{R} \boldsymbol{\omega})\text{tr}\left(\mathbf{\Sigma} \tilde{\mathbf{P}}^\top \tilde{\mathbf{P}}\right).
\end{aligned}
\tag{40}
$$

Further, the gradients can be written as

$$
\nabla_{\tilde{\mathbf{P}}}\mathcal{L}(\tilde{\mathbf{P}}, \boldsymbol{\omega}) = -2\boldsymbol{\omega}^\top \mathbf{r} \mathbf{\Sigma} + 2M \|\boldsymbol{\omega}\|_{\ell_2}^2 \, \tilde{\mathbf{P}} + 2(\|\boldsymbol{\omega}\|_{\ell_2}^2 + \boldsymbol{\omega}^\top \mathbf{R} \boldsymbol{\omega})\mathbf{\Sigma} \tilde{\mathbf{P}},
\tag{41}
$$

$$
\nabla_{\boldsymbol{\omega}}\mathcal{L}(\tilde{\mathbf{P}}, \boldsymbol{\omega}) = -2\text{tr}\left(\mathbf{\Sigma} \tilde{\mathbf{P}}\right) \mathbf{r} + 2M\text{tr}\left(\tilde{\mathbf{P}}^\top \tilde{\mathbf{P}}\right) \boldsymbol{\omega} + 2\text{tr}\left(\mathbf{\Sigma} \tilde{\mathbf{P}}^\top \tilde{\mathbf{P}}\right)(\mathbf{I}_n + \mathbf{R})\boldsymbol{\omega}.
\tag{42}
$$

Using the first-order optimality condition, and setting $\nabla_{\tilde{\mathbf{P}}}\mathcal{L}(\tilde{\mathbf{P}}, \boldsymbol{\omega}) = 0$ and $\nabla_{\boldsymbol{\omega}}\mathcal{L}(\tilde{\mathbf{P}}, \boldsymbol{\omega}) = 0$, we obtain

$$
\begin{aligned}
\tilde{\mathbf{P}} &= \left(M \|\boldsymbol{\omega}\|_{\ell_2}^2 \, \mathbf{I} + (\|\boldsymbol{\omega}\|_{\ell_2}^2 + \boldsymbol{\omega}^\top \mathbf{R} \boldsymbol{\omega})\mathbf{\Sigma}\right)^{-1} \mathbf{\Sigma} \boldsymbol{\omega}^\top \mathbf{r} \\
&= \frac{\boldsymbol{\omega}^\top \mathbf{r}}{M \|\boldsymbol{\omega}\|_{\ell_2}^2} \left(\frac{\|\boldsymbol{\omega}\|_{\ell_2}^2 + \boldsymbol{\omega}^\top \mathbf{R} \boldsymbol{\omega}}{M \|\boldsymbol{\omega}\|_{\ell_2}^2}\mathbf{I} + \mathbf{\Sigma}^{-1}\right)^{-1} \\
&= \frac{\boldsymbol{\omega}^\top \mathbf{r}}{M \|\boldsymbol{\omega}\|_{\ell_2}^2} \left(\frac{\gamma + 1}{M} \cdot \mathbf{I} + \mathbf{\Sigma}^{-1}\right)^{-1},
\end{aligned}
\tag{43a}
$$

where $\gamma = \boldsymbol{\omega}^\top \mathbf{R} \boldsymbol{\omega} / \|\boldsymbol{\omega}\|_{\ell_2}^2$.

Further,

$$
\begin{aligned}
\boldsymbol{\omega} &= \left(\left(M\text{tr}\left(\tilde{\mathbf{P}}^\top \tilde{\mathbf{P}}\right) + \text{tr}\left(\mathbf{\Sigma} \tilde{\mathbf{P}}^\top \tilde{\mathbf{P}}\right)\right)\mathbf{I} + \text{tr}\left(\mathbf{\Sigma} \tilde{\mathbf{P}}^\top \tilde{\mathbf{P}}\right)\mathbf{R}\right)^{-1} \text{tr}\left(\mathbf{\Sigma} \tilde{\mathbf{P}}\right) \mathbf{r} \\
&= \frac{\text{tr}\left(\mathbf{\Sigma} \tilde{\mathbf{P}}\right)}{M\text{tr}\left(\tilde{\mathbf{P}}^\top \tilde{\mathbf{P}}\right) + \text{tr}\left(\mathbf{\Sigma} \tilde{\mathbf{P}}^\top \tilde{\mathbf{P}}\right)} \left(\mathbf{I} + \frac{\text{tr}\left(\mathbf{\Sigma} \tilde{\mathbf{P}}^\top \tilde{\mathbf{P}}\right)}{M\text{tr}\left(\tilde{\mathbf{P}}^\top \tilde{\mathbf{P}}\right) + \text{tr}\left(\mathbf{\Sigma} \tilde{\mathbf{P}}^\top \tilde{\mathbf{P}}\right)}\mathbf{R}\right)^{-1} \mathbf{r}.
\end{aligned}
\tag{43b}
$$

Let

$$
\mathbf{\Sigma}_\gamma := \frac{\gamma + 1}{M} \cdot \mathbf{I} + \mathbf{\Sigma}^{-1}.
$$

Then, we get

$$
\begin{aligned}
\frac{\text{tr}\left(\mathbf{\Sigma} \tilde{\mathbf{P}}^\top \tilde{\mathbf{P}}\right)}{M\text{tr}\left(\tilde{\mathbf{P}}^\top \tilde{\mathbf{P}}\right) + \text{tr}\left(\mathbf{\Sigma} \tilde{\mathbf{P}}^\top \tilde{\mathbf{P}}\right)} &= \left(1 + M\frac{\text{tr}\left(\tilde{\mathbf{P}}^\top \tilde{\mathbf{P}}\right)}{\text{tr}\left(\mathbf{\Sigma} \tilde{\mathbf{P}}^\top \tilde{\mathbf{P}}\right)}\right)^{-1} \\
&= \left(1 + M\frac{\text{tr}\left(\mathbf{\Sigma}_\gamma^{-2}\right)}{\text{tr}\left(\mathbf{\Sigma} \mathbf{\Sigma}_\gamma^{-2}\right)}\right)^{-1} \\
&= \left(1 + M\sum_{i=1}^d \frac{s_i^2}{(M + (\gamma + 1)s_i)^2}\left(\sum_{i=1}^d \frac{s_i^3}{(M + (\gamma + 1)s_i)^2}\right)^{-1}\right)^{-1} \\
&=: g(\gamma).
\end{aligned}
$$

Here, the last equality follows from eigen decomposition $\mathbf{\Sigma} = \mathbf{U}\text{diag}(\mathbf{s})\mathbf{U}^\top$ with $\mathbf{s} = [s_1, \dots, s_d]^\top \in \mathbb{R}_{++}^d$.

Now, plugging $\tilde{P}$ defined in (43a) within $\omega$ given in (43b), we obtain

$$\omega = \frac{\text{tr}\left(\Sigma\tilde{P}\right)}{M\text{tr}\left(\tilde{P}^\top\tilde{P}\right) + \text{tr}\left(\Sigma\tilde{P}^\top\tilde{P}\right)} \cdot (g(\gamma)\cdot R + I)^{-1}\, r. \tag{44}$$

Using the above formulae for $\omega$, we rewrite $\gamma = \omega^\top R\omega/\|\omega\|_{\ell_2}^2$ as

$$\begin{aligned}
\gamma &= \frac{r^\top(g(\gamma)R + I)^{-1}R(g(\gamma)R + I)^{-1}r}{r^\top(g(\gamma)R + I)^{-2}r} \\
&= \sum_{i=1}^n \frac{\lambda_i a_i^2}{(1 + g(\gamma)\lambda_i)^2}\left(\sum_{i=1}^n \frac{a_i^2}{(1 + g(\gamma)\lambda_i)^2}\right)^{-1} \\
&=: h\left(g\left(\gamma\right)\right),
\end{aligned} \tag{45}$$

where the second equality follows from Assumption A and the fact that $R = E\text{diag}(\lambda)E^\top$ denotes the eigen decomposition of $R$, with $\lambda = [\lambda_1, \ldots, \lambda_n]^\top \in \mathbb{R}_+^n$.

Now, let $\gamma^\star$ denote a fixed point of composite function $h(g(\gamma))$. From (43a) and (44), we obtain

$$\tilde{P} = C(r, \omega, \Sigma)\cdot\left(\frac{\gamma^\star + 1}{M}\cdot I + \Sigma^{-1}\right)^{-1}, \quad \text{and}$$
$$\omega = c(r, \omega, \Sigma)\cdot\left(g(\gamma^\star)\cdot R + I\right)^{-1}\, r. \tag{46}$$

for some $C(r, \omega, \Sigma) = \frac{\omega^\top r}{M\|\omega\|_{\ell_2}^2}$ and $c(r, \omega, \Sigma) = \frac{\text{tr}(\Sigma\tilde{P})}{M\text{tr}(\tilde{P}^\top\tilde{P}) + \text{tr}(\Sigma\tilde{P}^\top\tilde{P})}$.

Now, using the our definition $\tilde{P} = \sqrt{\Sigma}P\sqrt{\Sigma}$, we obtain

$$P(\gamma) = C(r, \omega, \Sigma)\cdot\Sigma^{-\frac{1}{2}}\left(\frac{\gamma^\star + 1}{\sigma^2 + \text{tr}(\Sigma)}\cdot\Sigma + I\right)^{-1}\Sigma^{-\frac{1}{2}}, \quad \text{and}$$
$$\omega(\gamma) = c(r, \omega, \Sigma)\cdot\left(g(\gamma^\star)\cdot R + I\right)^{-1}r.$$

This completes the proof. $\qquad\qquad\square$

### C.2 Proof of Theorem 4

We first provide the following Lemma.

**Lemma 1.** *Let the functions $h : \mathbb{R}_+ \to \mathbb{R}_+$ and $g : \mathbb{R}_+ \to \mathbb{R}_+$ be defined as*

$$h(\bar{\gamma}) = \sum_{i=1}^n \frac{\lambda_i a_i^2}{(1 + \bar{\gamma}\lambda_i)^2}\left(\sum_{i=1}^n \frac{a_i^2}{(1 + \bar{\gamma}\lambda_i)^2}\right)^{-1}, \tag{47}$$

$$g(\gamma) = \left(1 + M\sum_{i=1}^d \frac{s_i^2}{(M + (\gamma + 1)s_i)^2}\left(\sum_{i=1}^d \frac{s_i^3}{(M + (\gamma + 1)s_i)^2}\right)^{-1}\right)^{-1}, \tag{48}$$

*where $M = \sigma^2 + \sum_{i=1}^d s_i$.*

*Suppose $\Delta_\Sigma \cdot \Delta_R < M + s_{\min}$, where $\Delta_\Sigma$ and $\Delta_R$ denote the effective spectral gaps of $\Sigma$ and $R$, respectively, as given in (12); and $s_{\min}$ is the smallest eigenvalue of $\Sigma$. We have that*

$$\left|\frac{\partial g}{\partial\gamma}\cdot\frac{\partial h}{\partial g}\right| \le \frac{\Delta_\Sigma^2 \cdot \Delta_R^2}{(M + s_{\min})^2} < 1.$$

*Proof.* Let

$$B(\gamma) = \sum_{i=1}^d \frac{s_i^3}{(M + (\gamma + 1)s_i)^2}, \quad C(\gamma) = \sum_{i=1}^d \frac{s_i^2}{(M + (\gamma + 1)s_i)^2}, \quad A(\gamma) = 1 + M\frac{C(\gamma)}{B(\gamma)}.$$

The derivatives of $B(\gamma)$ and $C(\gamma)$ are

$$B'(\gamma) = -2 \sum_{i=1}^{d} \frac{s_i^4}{(M + (\gamma + 1)s_i)^3}, \quad C'(\gamma) = -2 \sum_{i=1}^{d} \frac{s_i^3}{(M + (\gamma + 1)s_i)^3}.$$

The gradient of $g(\gamma)$ is

$$\frac{\partial g}{\partial \gamma} = -M \left( \frac{1}{A(\gamma)B(\gamma)} \right)^2 (C'(\gamma)B(\gamma) - C(\gamma)B'(\gamma)). \tag{49}$$

It can be seen that

$$\left( \frac{1}{A(\gamma)} \right)^2 \le M^{-2} \left( \sum_{i=1}^{d} \frac{s_i^3}{(M + (\gamma + 1)s_i)^2} \right)^2 \left( \sum_{i=1}^{d} \frac{s_i^2}{(M + (\gamma + 1)s_i)^2} \right)^{-2},$$

which implies that

$$\left( \frac{1}{A(\gamma)B(\gamma)} \right)^2 \le M^{-2} \left( \sum_{i=1}^{d} \frac{s_i^2}{(M + (\gamma + 1)s_i)^2} \right)^{-2}. \tag{50a}$$

Further, we have

$$\begin{aligned}
C'(\gamma)B(\gamma) - C(\gamma)B'(\gamma) &= -2 \sum_{i=1}^{d} \frac{s_i^3}{(M + (\gamma + 1)s_i)^3} \sum_{i=1}^{d} \frac{s_i^3}{(M + (\gamma + 1)s_i)^2} \\
&\quad + 2 \sum_{i=1}^{d} \frac{s_i^2}{(M + (\gamma + 1)s_i)^2} \sum_{i=1}^{d} \frac{s_i^4}{(M + (\gamma + 1)s_i)^3} \\
&= \sum_{i=1}^{d} \sum_{j=1}^{d} \frac{T_{ij}}{(M + (\gamma + 1)s_i)^3 (M + (\gamma + 1)s_j)^3} \\
&= M \cdot \sum_{i=1}^{d} \sum_{j=1}^{d} \frac{s_i^2 s_j^2 (s_i - s_i)^2}{(M + (\gamma + 1)s_i)^3 (M + (\gamma + 1)s_j)^3},
\end{aligned} \tag{50b}$$

where

$$\begin{aligned}
T_{ij} &= s_i^2 (M + (\gamma + 1)s_i)s_j^4 + s_i^4 s_j^2 (M + (\gamma + 1)s_j) \\
&\quad - s_i^3 s_j^3 (M + (\gamma + 1)s_j) - s_i^3 (M + (\gamma + 1)s_i)s_j^3 \\
&= s_i^2 s_j^2 \left( M \cdot (s_j^2 + s_i^2 - 2s_i s_j) + (\gamma + 1)(s_i s_j^2 + s_i^2 s_j - s_i s_j^2 - s_i^2 s_j) \right).
\end{aligned} \tag{50c}$$

Thus, substituting (50a) and (50b) into (49), we obtain

$$\left| \frac{\partial g}{\partial \gamma} \right| \le M \cdot M^{-1} \left( \sum_{i=1}^{d} \frac{s_i^2}{(M + (\gamma + 1)s_i)^2} \right)^{-2} \sum_{i,j=1}^{d} \frac{s_i^2 s_j^2 (s_i - s_j)^2}{(M + (\gamma + 1)s_i)^3 (M + (\gamma + 1)s_j)^3}. \tag{51}$$

Next, we derive $\frac{\partial h}{\partial \bar{\gamma}}$. Let

$$D(\bar{\gamma}) = \sum_{i=1}^{n} \frac{\lambda_i a_i^2}{(1 + \bar{\gamma} \lambda_i)^2}, \quad E(\bar{\gamma}) = \sum_{i=1}^{n} \frac{a_i^2}{(1 + \bar{\gamma} \lambda_i)^2}.$$

We have

$$D'(\bar{\gamma}) = -2 \sum_{i=1}^{n} \frac{\lambda_i^2 a_i^2}{(1 + \bar{\gamma} \lambda_i)^3}, \quad E'(\bar{\gamma}) = -2 \sum_{i=1}^{n} \frac{\lambda_i a_i^2}{(1 + \bar{\gamma} \lambda_i)^3}.$$

The derivative of $h$ with respect to $\bar{\gamma}$ is given by

$$\frac{\partial h}{\partial \bar{\gamma}} = - \left( \frac{1}{E(\bar{\gamma})} \right)^2 (E(\bar{\gamma})D'(\bar{\gamma}) - D(\bar{\gamma})E'(\bar{\gamma})). \tag{52}$$

Substituting into (52), we get

$$\left(\frac{1}{E(\bar{\gamma})}\right)^2 = \left(\sum_{i=1}^{n} \frac{a_i^2}{(1+\bar{\gamma}\lambda_i)^2}\right)^{-2}, \tag{53a}$$

and

$$E(\bar{\gamma})D'(\bar{\gamma}) - D(\bar{\gamma})E'(\bar{\gamma}) = 2\sum_{i=1}^{n} \frac{\lambda_i^2 a_i^2}{(1+\bar{\gamma}\lambda_i)^3} \sum_{i=1}^{n} \frac{a_i^2}{(1+\bar{\gamma}\lambda_i)^2}$$

$$- 2\sum_{i=1}^{n} \frac{\lambda_i a_i^2}{(1+\bar{\gamma}\lambda_i)^2} \sum_{i=1}^{n} \frac{a_i^2 \lambda_i}{(1+\bar{\gamma}\lambda_i)^3}$$

$$= \sum_{i=1}^{n} \sum_{j=1}^{n} \frac{\bar{T}_{ij}}{(1+\bar{\gamma}\lambda_i)^3(1+\bar{\gamma}\lambda_j)^3} \tag{53b}$$

$$= \sum_{i=1}^{n} \sum_{j=1}^{n} \frac{a_i^2 a_j^2 \left(\lambda_i^2 + \lambda_j^2 - 2\lambda_i\lambda_j\right)}{(1+\bar{\gamma}\lambda_i)^3(1+\bar{\gamma}\lambda_j)^3}.$$

Here,

$$\bar{T}_{ij} = \lambda_i^2 a_i^2 a_j^2 (1+\bar{\gamma}\lambda_j) + a_i^2(1+\bar{\gamma}\lambda_i)\lambda_j^2 a_j^2$$

$$- \lambda_i a_i^2(1+\bar{\gamma}\lambda_i)a_j^2\lambda_j - a_i^2\lambda_i\lambda_j a_j^2(1+\bar{\gamma}\lambda_j) \tag{53c}$$

$$= a_i^2 a_j^2 \left(\lambda_i^2(1+\bar{\gamma}\lambda_j) + (1+\bar{\gamma}\lambda_i)\lambda_j^2 - \lambda_i(1+\bar{\gamma}\lambda_i)\lambda_j - \lambda_i\lambda_j(1+\bar{\gamma}\lambda_j)\right).$$

Hence, substituting (53a) and (53b) into (52) gives

$$\frac{\partial h}{\partial \bar{\gamma}} = -\left(\sum_{i=1}^{n} \frac{a_i^2}{(1+\bar{\gamma}\lambda_i)^2}\right)^{-2} \sum_{i,j=1}^{n} \frac{a_i^2 a_j^2 (\lambda_i - \lambda_j)^2}{(1+\bar{\gamma}\lambda_i)^3(1+\bar{\gamma}\lambda_j)^3}. \tag{54}$$

Now, for the combined derivative, we have

$$\left|\frac{\partial g}{\partial \gamma} \cdot \frac{\partial h}{\partial \bar{\gamma}}\right| \le \left(\sum_{i=1}^{d} \frac{s_i^2}{(M+(\gamma+1)s_i)^2}\right)^{-2} \sum_{i,j=1}^{d} \frac{s_i^2 s_j^2 (s_i - s_j)^2}{(M+(\gamma+1)s_i)^3 \left(M+(\gamma+1)s_j\right)^3}$$

$$\cdot \left(\sum_{i=1}^{n} \frac{a_i^2}{(1+\bar{\gamma}\lambda_i)^2}\right)^{-2} \sum_{i,j=1}^{n} \frac{a_i^2 a_j^2 (\lambda_i - \lambda_j)^2}{(1+\bar{\gamma}\lambda_i)^3(1+\bar{\gamma}\lambda_j)^3}.$$

Note that $M+(\gamma+1)s_i$ and $1+\bar{\gamma}\lambda_j$ are nonnegative for all $i, j$. Hence,

$$\left|\frac{\partial g}{\partial \gamma} \cdot \frac{\partial h}{\partial \bar{\gamma}}\right| \le \left(\sum_{i=1}^{d} \frac{s_i^2 \left(M+(\gamma+1)s_i\right)}{(M+(\gamma+1)s_i)^3}\right)^{-2}$$

$$\cdot \sum_{i,j=1}^{d} \frac{s_i^2 s_j^2 \cdot \Delta_1 \cdot \left(M+(\gamma+1)s_j\right)\left(M+(\gamma+1)s_i\right)}{(M+(\gamma+1)s_i)^3 \left(M+(\gamma+1)s_j\right)^3}$$

$$\cdot \left(\sum_{i=1}^{n} \frac{a_i^2(1+\bar{\gamma}\lambda_i)}{(1+\bar{\gamma}\lambda_i)^3}\right)^{-2}$$

$$\cdot \sum_{i,j\in\mathcal{S}} \frac{a_i^2 a_j^2 \cdot \Delta_2 \cdot (1+\bar{\gamma}\lambda_i)\left(1+\bar{\gamma}\lambda_j\right)}{(1+\bar{\gamma}\lambda_i)^3 \left(1+\bar{\gamma}\lambda_j\right)^3},$$

where

$$\Delta_1 := \max_{i,j} \frac{(s_i - s_j)^2}{\left(M+(\gamma+1)s_j\right)\left(M+(\gamma+1)s_i\right)}, \quad \Delta_2 := \max_{i,j\in\mathcal{S}} \frac{(\lambda_i - \lambda_j)^2}{(1+\bar{\gamma}\lambda_i)(1+\bar{\gamma}\lambda_j)}. \tag{55}$$

Here, $\mathcal{S} = \{i \in [n] | \lambda_i \neq 0\} \subset [n]$.

Finally, setting $\bar{\gamma} = g(\gamma)$, we obtain

$$\left| h'(g(\gamma)) \cdot g'(\gamma) \right| = \left| \frac{\partial g}{\partial \gamma} \cdot \frac{\partial h}{\partial g} \right| \leq \Delta_1 \cdot \Delta_2 \leq \frac{\Delta_{\boldsymbol{\Sigma}}^2 \cdot \Delta_{\boldsymbol{R}}^2}{(M + s_{\min})^2} < 1.$$

where $\Delta_{\boldsymbol{\Sigma}}$ and $\Delta_{\boldsymbol{R}}$ are the spectral gaps of $\boldsymbol{\Sigma}$ and $\boldsymbol{R}$; and $s_{\min}$ is the smallest eigenvalue of $\boldsymbol{\Sigma}$; and $M = \sigma^2 + \sum_{i=1}^d s_i$. $\qquad \square$

*Proof of Theorem 4.* Lemma 1 shows that $|\partial h(g(\gamma))/\partial \gamma| < 1$, and as a result, the mapping $h(g(\gamma))$ on $\mathbb{R}_+$ is a contraction mapping. Therefore, by the Banach fixed-point theorem, this guarantees the existence of a unique root, denoted as $\gamma = \gamma^\star$. This completes the proof of **T1**. In the following, we provide the proof of **T2**. Substituting the unique $\gamma^\star$ into (16) and using the fact that $\tilde{\boldsymbol{P}} = \sqrt{\boldsymbol{\Sigma}} \boldsymbol{P} \sqrt{\boldsymbol{\Sigma}}$, we obtain $(\boldsymbol{P}^\star, \boldsymbol{\omega}^\star)$, as given in (14), as a global minima of (3).

Next, we claim that $(\boldsymbol{P}^\star, \boldsymbol{\omega}^\star)$ is the unique global minimizer of $\mathcal{L}(\boldsymbol{P}, \boldsymbol{\omega})$ up to rescaling, i.e., any other minimizer $(\hat{\boldsymbol{P}}, \hat{\boldsymbol{\omega}})$ must be related to $(\boldsymbol{P}^\star, \boldsymbol{\omega}^\star)$ by scaling factors $\alpha$ and $\beta$, such that $\hat{\boldsymbol{P}} = \alpha \boldsymbol{P}^\star$ and $\hat{\boldsymbol{\omega}} = \beta \boldsymbol{\omega}^\star$, for some $\alpha, \beta > 0$.

The loss function is given by

$$\mathcal{L}(\boldsymbol{P}, \boldsymbol{\omega}) = M - 2\mathrm{tr}\left(\boldsymbol{\Sigma}^2 \boldsymbol{P}\right) \boldsymbol{\omega}^\top \boldsymbol{r} + M \|\boldsymbol{\omega}\|^2 \mathrm{tr}\left(\boldsymbol{\Sigma} \boldsymbol{P}^\top \boldsymbol{\Sigma} \boldsymbol{P}\right) + (\|\boldsymbol{\omega}\|^2 + \boldsymbol{\omega}^\top \boldsymbol{R} \boldsymbol{\omega}) \mathrm{tr}\left(\boldsymbol{\Sigma}^2 \boldsymbol{P}^\top \boldsymbol{\Sigma} \boldsymbol{P}\right)$$

Now, consider the effect of rescaling the variables $\boldsymbol{P}$ and $\boldsymbol{\omega}$ by introducing scalars $\alpha$ and $\beta$, i.e., we substitute $\alpha \boldsymbol{P}$ and $\beta \boldsymbol{\omega}$ into the loss function

$$\mathcal{L}(\alpha \boldsymbol{P}, \beta \boldsymbol{\omega}) = M - 2\alpha\beta \mathrm{tr}(\boldsymbol{\Sigma}^2 \boldsymbol{P}) \boldsymbol{\omega}^\top \boldsymbol{r} + M\alpha^2\beta^2 \|\boldsymbol{\omega}\|^2 \mathrm{tr}(\boldsymbol{\Sigma} \boldsymbol{P}^\top \boldsymbol{\Sigma} \boldsymbol{P}) + \alpha^2\beta^2 (\|\boldsymbol{\omega}\|^2 + \boldsymbol{\omega}^\top \boldsymbol{R} \boldsymbol{\omega}) \mathrm{tr}(\boldsymbol{\Sigma}^2 \boldsymbol{P}^\top \boldsymbol{\Sigma} \boldsymbol{P}).$$

Define

$$A := \mathrm{tr}(\boldsymbol{\Sigma}^2 \boldsymbol{P}) \boldsymbol{\omega}^\top \boldsymbol{r}, \quad B := \mathrm{tr}(\boldsymbol{\Sigma} \boldsymbol{P}^\top \boldsymbol{\Sigma} \boldsymbol{P}), \quad C := \|\boldsymbol{\omega}\|^2, \quad D := \boldsymbol{\omega}^\top \boldsymbol{R} \boldsymbol{\omega}, \quad E := \mathrm{tr}(\boldsymbol{\Sigma}^2 \boldsymbol{P}^\top \boldsymbol{\Sigma} \boldsymbol{P}).$$

Thus, the rescaled loss function becomes

$$\mathcal{L}(\alpha \boldsymbol{P}, \beta \boldsymbol{\omega}) = M - 2\alpha\beta A + M\alpha^2\beta^2 BC + \alpha^2\beta^2 (C + D)E.$$

For $(\boldsymbol{P}^\star, \boldsymbol{\omega}^\star)$ to be a minimizer, the partial derivatives of the loss function with respect to $\boldsymbol{P}$ and $\boldsymbol{\omega}$ must vanish at $(\boldsymbol{P}^\star, \boldsymbol{\omega}^\star)$. However, we consider the effect of the rescaling in terms of $\alpha$ and $\beta$. To find the stationary points of $\mathcal{L}(\alpha \boldsymbol{P}, \beta \boldsymbol{\omega})$, we differentiate with respect to $\alpha$ and $\beta$:

$$\frac{\partial \mathcal{L}}{\partial \alpha} = -2\beta A + 2M\alpha\beta^2 BC + 2\alpha\beta^2 (C + D)E,$$

$$\frac{\partial \mathcal{L}}{\partial \beta} = -2\alpha A + 2M\alpha^2\beta BC + 2\alpha^2\beta (C + D)E.$$

Setting these to zero, we obtain the system

$$\alpha\beta(MBC + (C + D)E) = A. \tag{56}$$

This condition must hold for any minimizer. Now, suppose there exists another minimizer $(\hat{\boldsymbol{P}}, \hat{\boldsymbol{\omega}})$ that also minimizes the loss function. By the first-order optimality conditions, $\alpha\beta$ must remain constant. This implies that any other minimizer $(\hat{\boldsymbol{P}}, \hat{\boldsymbol{\omega}})$ must be proportional to the original minimizer $(\boldsymbol{P}^\star, \boldsymbol{\omega}^\star)$, meaning

$$\hat{\boldsymbol{P}} = \alpha \boldsymbol{P}^\star \quad \text{and} \quad \hat{\boldsymbol{\omega}} = \beta \boldsymbol{\omega}^\star$$

for some scalars $\alpha, \beta > 0$ satisfying (56).

Thus, any global minimizer $(\hat{\boldsymbol{P}}, \hat{\boldsymbol{\omega}})$ is a *scaled* version of $(\boldsymbol{P}^\star, \boldsymbol{\omega}^\star)$, and no other distinct minimizer exists. This proves uniqueness up to rescaling. $\qquad \square$

### C.3 PROOF OF COROLLARY 2

*Proof.* Since by assumption $\boldsymbol{\Sigma} = \boldsymbol{I}$, it follows from (13b) that

$$g(\gamma^\star) = \left( 1 + (d + \sigma^2) \sum_{i=1}^d \frac{1}{(d + \sigma^2 + \gamma^\star + 1)^2} \left( \sum_{i=1}^d \frac{1}{(d + \sigma^2 + \gamma^\star + 1)^2} \right)^{-1} \right)^{-1}$$

$$= \frac{1}{d + \sigma^2 + 1}.$$

Substituting this into (14) gives

$$\boldsymbol{P}^\star = \boldsymbol{I}, \quad \text{and} \quad \boldsymbol{\omega}^\star = \left(\boldsymbol{R} + (d + \sigma^2 + 1)\boldsymbol{I}\right)^{-1} \boldsymbol{r}.$$

Now, recall that the objective function is given by

$$\mathcal{L}(\boldsymbol{\omega}) = M - 2\text{tr}(\boldsymbol{\Sigma}^2 \boldsymbol{P})\boldsymbol{\omega}^\top \boldsymbol{r} + M \|\boldsymbol{\omega}\|_{\ell_2} \text{tr}(\boldsymbol{\Sigma} \boldsymbol{P}^\top \boldsymbol{\Sigma} \boldsymbol{P}) + (\|\boldsymbol{\omega}\|^2 + \boldsymbol{\omega}^\top \boldsymbol{R} \boldsymbol{\omega})\text{tr}(\boldsymbol{\Sigma}^2 \boldsymbol{P}^\top \boldsymbol{\Sigma} \boldsymbol{P}),$$

and, by assumption, $M = \sigma^2 + d$.

Substituting $\boldsymbol{P}^\star = \boldsymbol{I}$ and $\boldsymbol{\omega}^\star = \left(\boldsymbol{R} + (d + \sigma^2 + 1)\boldsymbol{I}\right)^{-1} \boldsymbol{r}$ into the objective (38), and using $\boldsymbol{\Sigma} = \boldsymbol{I}$, we get:

$$\mathcal{L}(\boldsymbol{\omega}^\star) = (\sigma^2 + d) - 2 \cdot d \cdot \boldsymbol{r}^\top \boldsymbol{\omega}^\star + (\sigma^2 + d) \cdot \left\|\boldsymbol{\omega}^\star\right\|_{\ell_2}^2 d + d\left(\|\boldsymbol{\omega}^\star\|^2 + \boldsymbol{\omega}^\top \boldsymbol{R} \boldsymbol{\omega}^\star\right).$$

The expression simplifies as

$$\mathcal{L}(\boldsymbol{\omega}^\star) = (\sigma^2 + d) - 2d \cdot \boldsymbol{r}^\top \left(\boldsymbol{R} + (d + \sigma^2 + 1)\boldsymbol{I}\right)^{-1} \boldsymbol{r} + (\sigma^2 + d)d \left\|\boldsymbol{\omega}^\star\right\|_{\ell_2}^2 + d\left(\|\boldsymbol{\omega}^\star\|^2 + \boldsymbol{\omega}^{\star\top} \boldsymbol{R} \boldsymbol{\omega}^\star\right).$$

Next, we compute $\|\boldsymbol{\omega}^\star\|^2$ and $\boldsymbol{\omega}^\top \boldsymbol{R} \boldsymbol{\omega}^\star$. By definition, we have

$$\|\boldsymbol{\omega}^\star\|^2 = \boldsymbol{r}^\top \left(\boldsymbol{R} + (d + \sigma^2 + 1)\boldsymbol{I}\right)^{-2} \boldsymbol{r},$$

and

$$\boldsymbol{\omega}^{\star\top} \boldsymbol{R} \boldsymbol{\omega}^\star = \boldsymbol{r}^\top \left(\boldsymbol{R} + (d + \sigma^2 + 1)\boldsymbol{I}\right)^{-1} \boldsymbol{R} \left(\boldsymbol{R} + (d + \sigma^2 + 1)\boldsymbol{I}\right)^{-1} \boldsymbol{r}.$$

Thus,

$$(d + \sigma^2 + 1)\|\boldsymbol{\omega}^\star\|^2 + \boldsymbol{\omega}^{\star\top} \boldsymbol{R} \boldsymbol{\omega}^\star = \boldsymbol{r}^\top \left(\boldsymbol{R} + (d + \sigma^2 + 1)\boldsymbol{I}\right)^{-1} \left((d + \sigma^2 + 1)\boldsymbol{I} + \boldsymbol{R}\right)\left(\boldsymbol{R} + (d + \sigma^2 + 1)\boldsymbol{I}\right)^{-1} \boldsymbol{r}$$

$$= \boldsymbol{r}^\top \left(\boldsymbol{R} + (d + \sigma^2 + 1)\boldsymbol{I}\right)^{-1} \boldsymbol{r}.$$

Substituting this result back into the objective function gives

$$\mathcal{L}(\boldsymbol{\omega}^\star) = (\sigma^2 + d) - d \cdot \boldsymbol{r}^\top \left(\boldsymbol{R} + (d + \sigma^2 + 1)\boldsymbol{I}\right)^{-1} \boldsymbol{r}.$$

$\square$

# D  LOSS LANDSCAPE OF 1-LAYER GLA

## D.1  PROOF OF THEOREM 5

*Proof.* We first prove that under Assumption B, $\mathcal{L}_{\text{GLA}}^\star = \min_{\boldsymbol{P} \in \mathbb{R}^{d \times d}, \boldsymbol{\omega} \in \mathcal{W}} \mathcal{L}_{\text{WPGD}}(\boldsymbol{P}, \boldsymbol{\omega})$ where $\mathcal{W}$ is the search space of weighting vector $\boldsymbol{\omega} \in \mathbb{R}^n$ defined as

$$\mathcal{W} := \left\{ \left[\omega_1 \mathbf{1}_{n_1}^\top \; \cdots \; \omega_K \mathbf{1}_{n_K}^\top\right]^\top \in \mathbb{R}^n \;\middle|\; 0 \leq \omega_i \leq \omega_j \leq 1, \; \forall 1 \leq i \leq j \leq K \right\}.$$

Define a set $\bar{\mathcal{W}} := \left\{ [\omega_1 \; \cdots \; \omega_n]^\top \in \mathbb{R}^n \;\middle|\; 0 \leq \omega_i \leq \omega_j \leq 1, \; \forall 1 \leq i \leq j \leq n \right\}$ and we have $\mathcal{W} \in \bar{\mathcal{W}}$.

Given scalar gating $\boldsymbol{G}_i = \begin{bmatrix} * & * \\ g_i \mathbf{1}^\top & * \end{bmatrix}$, following (10), the weighting vector returns

$$\boldsymbol{\omega} := [g_{1:n+1} \; \cdots \; g_{n:n+1}]^\top.$$

Since GLA with scalar gating valued in $[0, 1]$ following Assumption B, that is, $g_i \in [0, 1]$. Therefore, we have $g_{i:n+1} \leq g_{j:n+1}$ for $1 \leq i \leq j \leq n$. Therefore, any weighting vector implemented by GLA gating should be inside $\bar{\mathcal{W}}$.

Next, we will show that

$$\boldsymbol{\omega}^\star \in \mathcal{W} \quad \text{where} \quad \boldsymbol{\omega}^\star = \arg \min_{\boldsymbol{P}, \boldsymbol{\omega} \in \bar{\mathcal{W}}} \mathcal{L}_{\text{WPGD}}(\boldsymbol{P}, \boldsymbol{\omega}).$$

Define the weighting vector $\boldsymbol{\omega} = [\boldsymbol{\omega}_1^\top \; \cdots \; \boldsymbol{\omega}_K^\top]^\top \in \mathbb{R}^n$ where we have $\boldsymbol{\omega}_k = [\omega_1^{(k)} \; \cdots \; \omega_{n_k}^{(k)}]^\top \in \mathbb{R}^{n_k}$. For any $\boldsymbol{\omega} \notin \mathcal{W}$, there exist $(i, j, k)$, $i = j - 1$ such that $\omega_i^{(k)} < \omega_j^{(k)}$. Given gradient in (42), we have that

$\nabla_{\omega_i^{(k)}} \mathcal{L} = c_1 \cdot \omega_i^{(k)} + c_2$ and $\nabla_{\omega_j^{(k)}} \mathcal{L} = c_1 \cdot \omega_j^{(k)} + c_2$ with for some $c_1, c_2$ with $c_1 > 0$. $\nabla_{\omega_i^{(k)}} \mathcal{L} < \nabla_{\omega_j^{(k)}} \mathcal{L}$.
Therefore either increasing $\omega_j^{(k)}$ (if $\nabla_{\omega_i^{(k)}} \mathcal{L} < 0$) or decreasing $\omega_j^{(k)}$ (if $\nabla_{\omega_j^{(k)}} \mathcal{L} > 0$) will reduce the loss.
This results in showing that $\omega^\star \in \mathcal{W}$.

Finally, we will show that any $\omega \in \mathcal{W}$ can be obtained via the GLA gating. Let $\omega = [\omega_1 \mathbf{1}_{n_1}^\top \cdots \omega_K \mathbf{1}_{n_K}^\top]^\top$ be any vector in $\mathcal{W}$ and assume that $\omega_K = \alpha < 1$ without loss of generality. Then such sample weighting can be achieved via the gating

$$\begin{bmatrix} \mathbf{1}_{n_1}^\top & \frac{\omega_1}{\omega_{1:K}} & \cdots & \mathbf{1}_{n_k}^\top & \frac{\omega_k}{\omega_{k:K}} & \cdots & \mathbf{1}_{n_K}^\top & \frac{\omega_K}{\omega_{K:K}} \end{bmatrix}^\top .$$

Let $\omega_k' := \frac{\omega_k}{\omega_{k:K}}$ and let $\boldsymbol{w}_g$ be in the form of

$$\boldsymbol{w}_g = \begin{bmatrix} \mathbf{0}_{d+1} \\ \tilde{\boldsymbol{w}}_g \end{bmatrix} \in \mathbb{R}^{d+p+1}.$$

Then it remains to show that there exists $\tilde{\boldsymbol{w}}_g$ satisfying:

$$\phi(\tilde{\boldsymbol{w}}_g^\top \bar{\boldsymbol{d}}_k) \begin{cases} = 1, & k = 0 \\ = \omega_k', & k \in [K] \end{cases}$$

Assumption B implies the feasible of the problem, which completes the proof of (23).

Recap the optimal weighting from (14) which takes the form of

$$\omega^\star = \left(g(\gamma^\star) \cdot \boldsymbol{R} + \boldsymbol{I}\right)^{-1} \boldsymbol{r}.$$

Since Assumption C holds and $n_1 = n_2 = \cdots = n_K := \bar{n}$, $\omega^\star$ takes the form of $\omega^\star = c\boldsymbol{r}$ for some positive constant $c$. Therefore, the optimal weighting (up to a scalar) is inside the set $\mathcal{W}$. Combining it with (23) completes the proof.

$\square$

### D.2 Proof of Theorem 6

*Proof.* Following the similar proof of Theorem 5, and letting $\tilde{\mathcal{W}} := \left\{ \left[ \omega_1 \mathbf{1}_{n_1}^\top \cdots \omega_K \mathbf{1}_{n_K}^\top \right]^\top \in \mathbb{R}^n \right\}$, we obtain

$$\min_{\boldsymbol{P}, \omega \in \tilde{\mathcal{W}}} \mathcal{L}_{\text{WPGD}}(\boldsymbol{P}, \omega) = \min_{\boldsymbol{P}, \omega} \mathcal{L}_{\text{WPGD}}(\boldsymbol{P}, \omega).$$

Therefore, it remains to show that any $\omega \in \tilde{\mathcal{W}}$ can be implemented via some gating function. Let $\omega = [\omega_1 \mathbf{1}_{n_1}^\top \cdots \omega_K \mathbf{1}_{n_K}^\top]$ be arbitrary weighting in $\tilde{\mathcal{W}}$. Theorem 5 has shown that if $\omega_1 \leq \omega_2 \leq \cdots \leq \omega_K$, GLA with scalar function can implement such increasing weighting.

Now, inspired from Appendix B that all dimensions in the output implement individual WPGD. We can decouple the weighting into $K$ separate weighting:

$$\begin{aligned}
\omega_1 &= \omega_1 [\mathbf{1}_{n_1}^\top \cdots \mathbf{1}_{n_K}^\top] \\
\omega_2 &= (\omega_2 - \omega_1)[\mathbf{0}_{n_1}^\top \ \mathbf{1}_{n_2}^\top \cdots \mathbf{1}_{n_K}^\top] \\
\omega_3 &= (\omega_3 - \omega_2)[\mathbf{0}_{n_1}^\top \ \mathbf{0}_{n_2}^\top \ \mathbf{1}_{n_3}^\top \cdots \mathbf{1}_{n_K}^\top] \\
&\cdots \\
\omega_K &= (\omega_3 - \omega_2)[\mathbf{0}_{n_1}^\top \ \mathbf{0}_{n_2}^\top \ \mathbf{0}_{n_3}^\top \cdots \mathbf{0}_{n_{K-1}}^\top \ \mathbf{1}_{n_K}^\top]
\end{aligned}$$

and we have $\omega = \sum_{k=1}^K \omega_k$. Recap from Appendix B and consider the construction $\boldsymbol{W}_v = [\mathbf{0}_{(d+1) \times d} \ \boldsymbol{u}]$. Assumption B implies that $K \leq p < d + p + 1$.

From (27) and (28), let $i$'th dimension implements the weighting $\omega_i$ for $i \in [K]$. Specifically, let $g^i$ implement weighting $[\mathbf{0}_{n_1}^\top \cdots \mathbf{0}_{n_{i-1}}^\top \ \mathbf{1}_{n_i}^\top \cdots \mathbf{1}_{n_k}^\top]$ (which is feasible due to Theorem 5) and set $u_i = \omega_i - \omega_j$ with $\omega_0 = 0$. Then the composed weighting following (28) returns $\omega$, which completes the proof. $\square$

### D.3 PROOF OF COROLLARY 3

*Proof.* Recap from (43a) that given $\mathbf{\Sigma} = \boldsymbol{I}$ and $\boldsymbol{\omega} = \mathbf{1}$,

$$\boldsymbol{P}^\star = \left((d + \sigma^2)n\boldsymbol{I} + (n + \mathbf{1}^\top \boldsymbol{R}\mathbf{1})\boldsymbol{I}\right)^{-1} \mathbf{1}^\top \boldsymbol{r}$$

$$= \frac{\mathbf{1}^\top \boldsymbol{r}}{n(d + \sigma^2 + 1) + \mathbf{1}^\top \boldsymbol{R}\mathbf{1}}\boldsymbol{I} := c\boldsymbol{I}.$$

Then taking it back to the loss function (c.f. (38)) obtains

$$\mathcal{L}(\boldsymbol{P}^\star, \boldsymbol{\omega} = \mathbf{1}) = d + \sigma^2 - 2cd\mathbf{1}^\top \boldsymbol{r} + (d + \sigma^2)c^2 nd + (n + \mathbf{1}^\top \boldsymbol{R}\mathbf{1})c^2 d$$

$$= d + \sigma^2 - cd\mathbf{1}^\top \boldsymbol{r}.$$

It completes the proof. $\square$

