# OpenReview forum: "Gating is Weighting: Understanding Gated Linear Attention through In-context Learning"
_ICLR.cc/2025/Conference — Submitted to ICLR 2025_

### Official Review · Reviewer_xUCq · 2024-11-02

**Soundness:** 3
**Presentation:** 3
**Contribution:** 3
**Rating:** 6
**Confidence:** 2

**Summary:**

This work establishes a connection between Gated Linear Attention (GLA) architectures and Weighted Projected Gradient Descent (WPGD) algorithms with data-dependent weights in the context of in-context learning (ICL). It characterizes the optimization landscape of learning a WPGD algorithm and then studies the optimization landscape of GLA.

**Strengths:**

The paper is well-written and easy to follow.

It addresses a scenario of ICL in which training data are derived from multiple tasks, offering a more realistic framework than those considered in prior works.

**Weaknesses:**

The architecture considered in this paper appears to include only the GLA layer and no MLP layer. Consequently, a multi-layer GLA in section 3.2 would not fully align with a Transformer model. A discussion on the effects of the MLP layer would provide valuable insights.

In Section 5, the multitask setup appears to be simplified through the introduction of vectors $d$ as task boundaries and vectors $c$ as contextual features. Could you clarify the effect of each of $c$ and $d$ on optimal loss in this setting? Additionally, it would be insightful to evaluate the performance impact of including versus excluding each of $c$ and $d$ when training on real data.

Minor Comments
- Typo on line 179: "weight" should be "weigh".
- The label "n" of the x-axis in Figure 1 should be clarified for better understanding.

**Questions:**

Please see above.

---

> ### Author Response · Authors · 2024-11-23
>
> > **Weakness 1:** The architecture considered in this paper appears to include only the GLA layer and no MLP layer. Consequently, a multi-layer GLA in section 3.2 would not fully align with a Transformer model. A discussion on the effects of the MLP layer would provide valuable insights.
>
>
> **Response:** We appreciate the reviewer’s suggestion. The architecture in this paper includes the GLA layer without incorporating an MLP layer. Following your valuable feedback, we have added the following paragraph below Theorem 2 in Section 3.2:
>
>
>     Our theoretical results in Theorem 2 focus on multi-layer GLA without Multi-Layer Perceptron (MLP) layers to isolate and analyze the effects of the gating mechanism. However, MLP layers, a key component of standard Transformers, enable deeper feature transformations and non-linear interactions, potentially enhancing GLA's expressive power. Future work could explore the theoretical foundations of integrating MLPs into GLA and analyze the optimization landscape of general gated attention models, aligning them more closely with conventional Transformer architectures (Gu & Dao, 2023; Dao & Gu, 2024; Peng et al., 2024).
>
> Please refer to the first paragraph (highlighted) after Corollary 1.
>
>
> > **Weakness 2:** In Section 5, the multitask setup appears to be simplified through the introduction of vectors $d$ as task boundaries and vectors $c$ as contextual features. Could you clarify the effect of each of $c$ and $d$ on optimal loss in this setting? Additionally, it would be insightful to evaluate the performance impact of including versus excluding each of $c$ and $d$ when training on real data.
>
> **Response:** Thank you for the insightful question regarding the role of contextual features in Section 5. Here is our detailed response:
>
> **Theoretical and empirical perspective:**
>  - As shown in Theorem 1, the weighting matrix $\Omega$ is highly complex, as it depends heavily on the non-linear gating function and the input embeddings. To ensure that the optimal weighting induced by gating remains predictable, we introduced contextual features $c$ and $d$.
>
> - The green curves in Figure 1 illustrate that, without delimiters, a one-layer GLA cannot achieve optimal performance, as demonstrated by the theoretical predictions (black dashed curves). However, with delimiters, the optimal loss is achievable, as shown in Figure 1(a-c), where red and black curves overlap.
>
> - Theorem 5 demonstrates that when $c$ and $d$'s are **linearly independent**, the choice of these contextual features does not affect the optimal loss, and any valid contextual features will result in the same loss. However, when $c$ and $d$ are not linearly independent, their influence becomes significant. In such cases, better optimization can be achieved by assigning relevant contextual features to related tasks and distinct contextual features to unrelated tasks.
>
> **Practical perspective on real data:** Real data often includes inherent structures like verbs, objects, and nouns in language data. Proper embeddings allow models to categorize and distinguish these elements into different tasks effectively. However, in our setting, the input $z \in Z$ consists of random features $x \sim \mathcal{N}(0, I)$ without task-specific information. In such scenarios, without contextual features, the model struggles to distinguish between tasks based solely on the random input. As demonstrated empirically in Figure 1 (green curves), excluding contextual features leads to significantly degraded predictions by the GLA model.
> Additionally, contextual features have been utilized in prior works such as [Wang et al. (2024)](https://arxiv.org/pdf/2407.00256), [Asai et al. (2022)](https://arxiv.org/pdf/2205.11961), and [Dun et al. (2023)](https://arxiv.org/pdf/2310.02842). These works demonstrate the practical significance of contextual features in handling multitask setups. We have expanded the discussion in the paper to include these insights and references.
>
> > **Weakness 3:** Minor comments
>
> **Response:** Fixed. Thank you!

---

> > ### Comment · Reviewer_xUCq · 2024-11-26
> > **Response to Rebuttal**
> >
> > Thank you for your response. My score remains unchanged.

---

### Official Review · Reviewer_Y6dU · 2024-11-03

**Soundness:** 4
**Presentation:** 3
**Contribution:** 3
**Rating:** 6
**Confidence:** 3

**Summary:**

This paper tries to formulate and explore Gated Linear Attention (GLA) models (e.g. Mamba, RWKV) through in-context learning.
The main contributions are:
* Demonstrating that GLA models can implement data-dependent Weighted Projected Gradient Descent (WPGD) algorithms, where weights are induced by the gating function.
* Investigating the problem of learning an optimal WPGD, and by characterizing the loss landscape under a multitask data setting, showing the conditions under which there exists a unique global minimum.
* Characterizing the loss landscape of a 1-layer GLA and showing the constraints on the optimal weights.
*  Showing the differences between linear attention and GLA (with scaler and vector gating) and show scalar gating can has limited expressivity which can be enhanced by vector gating.

**Strengths:**

* I commend the authors' effort to establish a theoretical foundation for understanding GLA models.

* As the authors cited, the closest work to this work is Li et al. (2024). Using in-context learning, they also showed that H3 architectures (which are also GLA) can implement Weighted Projected Gradient Descent. So, this new paper has incremental contributions such as generalizing the formulation to more complicated GLA models by introducing a novel data model with non-IID examples and multitask prompts.

* Comparing the effects of scalar and vector gating mechanisms on performance provides valuable insights for crafting models.

**Weaknesses:**

* The paper has strong theoretical contributions. But, more empirical studies and comparisons could strengthen the practical applicability of the theoretical contributions.

* There exists another line of work (which is not based on in-context learning) such as papers cited below. They propose an implicit self-attention formulation for GLA models (including Mamba and RWKV). Do you think there is a connection between your work and this line of work, and is it possible to apply in-context learning for model explainablity and empirical studies?

Zong C, Shao J, Lu W, Zhuang Y. Stock Movement Prediction with Multimodal Stable Fusion via Gated Cross-Attention Mechanism. arXiv preprint arXiv:2406.06594. 2024 Jun 6.

Zimerman I, Ali A, Wolf L. A Unified Implicit Attention Formulation for Gated-Linear Recurrent Sequence Models. arXiv preprint arXiv:2405.16504. 2024 May 26.

**Questions:**

The same as above.

---

> ### Author Response · Authors · 2024-11-23
>
> > **Weakness 1:** The paper has strong theoretical contributions. But, more empirical studies and comparisons could strengthen the practical applicability of the theoretical contributions.
>
> **Response:** We thank the reviewer for acknowledging the strong theoretical foundation of our work. Regarding the suggestion for additional empirical studies, in Appendix A.1, we have extended our experiments to multi-layer GLA models and verified that deeper models achieve better predictions. Furthermore, numerous prior works (e.g., Table 1 in [Yang et al. (2024)](https://arxiv.org/pdf/2312.06635)) have successfully implemented GLA in real-world applications. While our study does not include real-application results, our theoretical contributions are both foundational and novel, and we believe our work provides significant contributions (as discussed in the General Response).
>
> > **Weakness 2:** There exists another line of work (which is not based on in-context learning) such as papers cited below. They propose an implicit self-attention formulation for GLA models (including Mamba and RWKV). Do you think there is a connection between your work and this line of work, and is it possible to apply in-context learning for model explainability and empirical studies?
>
> **Response:**  We appreciate the suggestion to explore connections with implicit self-attention frameworks.
>
>
>  [Zimmerman et al.  (2024)](https://arxiv.org/pdf/2405.16504)   propose a framework demonstrating how various architectures, including GLA models like Mamba and RWKV, can be viewed under a unified implicit attention perspective. This aligns with our theoretical exploration of GLA’s data-dependent Weighted Preconditioned Gradient Descent (WPGD), as both approaches emphasize data-adaptive weights driven by gating mechanisms. We believe that the theoretical results in this paper, combined with the unified perspective of GLA as a variant of attention, offer a potential pathway for extending GLA’s analysis to the optimization landscapes of models like Mamba and RWKV and connecting them to WPGD.
>
> [Zong et al. (2024)](https://arxiv.org/pdf/2406.06594) leverage a gated cross-attention mechanism for robust multimodal fusion. Their approach emphasizes stable integration of heterogeneous data streams. While their task differs, the underlying gated mechanism aligns with GLA’s capacity to manage multi-task prompts by dynamically weighting inputs. This suggests that GLA’s gating mechanism can be repurposed for tasks beyond sequence modeling, including robust multimodal fusion.
> We have incorporated a summary of the above discussion into the Related Work section (Section 1.1)

---

> > ### Comment · Reviewer_Y6dU · 2024-11-23
> > **Response to Rebuttal**
> >
> > Thanks for the response. I keep my rating as is since I think the contribution is marginally above the acceptance threshold. I hope the AC makes an informed decisions by considering all factors.

---

### Official Review · Reviewer_41dD · 2024-11-03

**Soundness:** 3
**Presentation:** 2
**Contribution:** 2
**Rating:** 6
**Confidence:** 3

**Summary:**

This paper studies the In-context learning ability of gated linear attention (GLA) and the weighted projected gradient descent (WPGD). It first shows that the single layer of GLA can implement one step of WPGD, and multiple layer of GLA can implement multiple steps of WPGD. Then, it delves into the optimization landscape of WPGD and GLA. It first shows that under some conditions, there exists a stationary point of one-step WPGD when the tasks within a prompt are correlated. It also shows that this stationary point is the unique global minimum under some other conditions. Finally, it shows that under some (strong) assumptions, the optimal ICL risk of GLA matches the optimal ICL risk induced by WPGD. in general, I think this paper delves into a very interesting problem: how does GLA behave on ICL tasks (with correlated tasks), but I think the paper needs a good revision and I will lean towards a rejection.

**Strengths:**

1. This paper delves into a very interesting problem: how does GLA behave on ICl tasks with correlated tasks? Can it implement something other than one-step of gradient descent (like what linear self-attention does)?

2. The theory looks solid and the proof looks correct to my understanding. They also have experimental evidence to show the performance gap among scaler-gated, vector-gated, and vanilla linear self-attention.

**Weaknesses:**

1. They claim that they establish the 'equivalence' between WPGD and GLA layer (line 198-199), but their construction result only shows that the GLA can implement a special subset of the WPGD. This is not an 'equivalence' to my understanding. Also, in terms of the optimization results, you have proven that the global minimum of the ICL risk induced by GLA and WPGD matches under some very strong assumptions, which is far away from your claimed equivalence.

2. In the preliminary section, you introduce a token embedding matrix like (4), but this was not utilized in your submission. In the construction results in section 3, you show that GLA can implement WPGD using this token matrix, but for the optimization landscape result in section 5, you show the global minimum of the ICL risk of GLA estimator using a more complex token embedding matrix with delimiters (in equation 20). The setups in two sections are inconsistent and you did not show me why it is necessary to use the more complex token embedding matrix in terms of the optimization landscape. Is there any intrinsic drawback to use the original token matrix? Moreover, is there any practical case in real applications where people use a token embedding matrix like equation 20? I think the adoption of a more involved token embedding matrix in Equation 20 needs more verification.

3. You claim that you prove the advantage of using the gating scheme in linear self-attention, but you do not present any rigorous results showing that the ICL risks induced by the optimal linear self-attention (LSA) and the optimal GLA have large gaps and how large the gap is. You show the optimal ICL risks for LSA and GLA separately. Do these two optimal ICL risks hold under the exact same assumptions? If so, I will suggest to write a separate corollary to conclude how large the risk gap is and what magnitude it enjoys.

4. The assumptions in the submission needs more justification. For example, why do you need to assume  the delimiters are linearly independent? What is this activation function and what role  does it perform? In assumption C, why do you assume the correlation between tasks are zero for \beta_i and \beta_j (This is a very special case under definition 1, right?)

5. Theorem 6 seems confusing. In prior section, you are considering a scaler-gated linear self-attention and show that it can implement a PWGD estimator like equation (10) or equaion (3). The optimal ICL risk of PWGD is also established in this scaler-gated linear attention setup (like in Theorem 3 and 4). But in there 6, you claim that the optimal ICL risks induced by the vector-gated linear self-attention matches the optimal ICL risk of WPGD. I am wondering whether the L_{WPGD}^* in the theorem 6 means the optimal risk over the function class in equation 10? If so, there is a mismatch between these two setups, since you are saying that the optimal ICL risk induced by a wider class (vector-gated linear attention) can match the optimal ICL risk of WPGD class in equation 10, which a simpler class can induce. I think you should put more efforting in clearly stating the relationship among the scaler-gated, vector-gated linear attention, and a restricted subclass of WPGD represented by equation 10, as well as the optimal ICL risks induced by them.

**Questions:**

/

**Details Of Ethics Concerns:**

/

---

> ### Author Response · Authors · 2024-11-23
> **Response to Reviewer 41dD - Part I**
>
> > **Weakness 1:** They claim that they establish ... which is far away from your claimed equivalence.
>
> **Response:**  We appreciate the reviewer pointing out the ambiguity in our use of the term “equivalence”.
> - In Section 3.1, the equivalence refers to the fact that a one-layer GLA model under the specific construction implements one step of WPGD considering the weighting matrix $\Omega$ determined by the choice of gating function and input space. We have clarified this distinction in the revised manuscript to avoid misinterpretation.
> - In Section 5.2, the equivalence refers to the optimization prediction for GLA and WPGD being identical under certain data assumptions. This result aligns with the findings presented in Theorem 1 of Ahn et al. (2023) and Proposition 1 of Li et al. (2024). However, our work extends these analyses by considering a more complex model architecture, i.e., GLA, and data settings, i.e., task-mixture ICL.
>
> We kindly refer the reviewer to our response to **Weakness 4** for further justification regarding the "strong" assumptions.
>
>
> > **Weakness 2:** In the preliminary section, you introduce ... involved token embedding matrix in Equation 20 needs more verification.
>
> **Response:**  To clarify, we address each point of this comment sequentially:
> - The embedding matrix Eq. (4) is used in Sections 2 and 3, including Eq. (5), Theorem 1, and relevant portions of the text.
> - In Section 3, we demonstrate that a one-layer GLA performs one step of WPGD with the weighting matrix $\Omega$ determined by the gating function and input embedding. However, we do not claim that their optimization landscapes are the same. Analyzing the optimization of GLA with the token embedding in Eq. (4) is challenging because the weight matrix $\Omega$ heavily depends on the embedding (as per Theorem 1), and the embedding itself incorporates random features $x \sim \mathcal{N}(0, I)$. To further investigate GLA's optimization performance, we introduce the embedding matrix in Eq. (20) with additional delimiters in Section 5. This ensures that the optimal data-dependent weighting is both analyzable and achievable. Furthermore, our empirical results show that without delimiters, the performance is non-smooth, non-optimal (as shown by the green curves in Figure 1), and cannot achieve the theoretically optimal performance (as indicated by the black dashed curves).
> - Regarding practical relevance, delimiters can be interpreted as "task transition" identifiers. For instance, delimiters/prompts have been used in the mixture-of-prompt literature, such as Eq. (1) in [Wang et al. (2024)](https://arxiv.org/pdf/2407.00256), Fig. 2 in [Asai et al. (2022)](https://arxiv.org/pdf/2205.11961), and Section 2 in [Dun et al. (2023)](https://arxiv.org/pdf/2310.02842), to separate different tasks. We appreciate the reviewer’s comment and have added this discussion to the paper.
>
>
> > **Weakness 3:** You claim that you prove ... how large the risk gap is and what magnitude it enjoys.
>
> **Response:** Thank you for suggesting a clearer comparison of the ICL risks for GLA and LSA. Based on your valuable feedback, we have added Corollary 4 to our paper, explicitly demonstrating that GLA outperforms linear attention in terms of ICL risk. This corollary now quantifies the risk gap and specifies the assumptions under which the comparison holds.

---

> > ### Author Response · Authors · 2024-11-23
> > **Response to Reviewer 41dD - Part II**
> >
> > > **Weakness 4:** The assumptions in the submission need more justification. ... (This is a very special case under definition 1, right?)
> >
> > **Response:**  To address the reviewer’s concerns regarding the assumptions, we have added more detailed justifications:
> > 1. **Assumption B (independence of delimiters):** The linear independence assumption ensures the feasibility of achieving all weightings in $\mathcal W$, as defined in Theorem 5. If delimiters $\bar d_1$ and $\bar d_2$ are linearly dependent, e.g., $\bar d_1=\bar d_2$, the gating outputs $\phi(w_g^\top d_1) = \phi(w_g^\top d_2)$. In this case, not all weighting vectors $\omega \in \mathcal W$ are achievable.
> > 2. **Assumption B (activation function):** The activation function $\phi$ is an element-wise non-linear function that maps the gating weights into a restricted range, such as $[0,1]$. This introduces non-linearity and enhances stability, preventing the weighting vectors from diverging. Examples include the sigmoid function (used in RWKV) and $\exp(-\text{softplus}(x))$ (used in Mamba). In our experiments, we utilize the sigmoid function.
> > 3. **Assumption C (correlation between tasks):** Assumption C ensures that the optimal weighting derived from Eq. (14) follows a non-decreasing order, ensuring the optimal weighting, $\omega^\star$, lies within the search space $\mathcal{W}$ defined in Theorem 5. Here is a counter-example without the zero-correlation assumption: Suppose
> > $$K=3,\ d=5,\ \sigma=0,\ R=\begin{bmatrix}1&-1&1\\\\-1&1&-1\\\\1&-1&1\end{bmatrix},\ \text{and}\ r=\begin{bmatrix}1\\\\1\\\\1\end{bmatrix}.$$
> > Following the data setup in Corollary 2, the optimal risk results in $\omega^\star\approx[0.148, 0.185, 0.148]^{\top}$, which lies outside the search space $\mathcal{W}$. However, this assumption is not strictly necessary and can be replaced by the sufficient and necessary condition: *The optimal weight $\omega^\star$ in Eq. (14) lies within $\mathcal{W}$.*
> >
> >
> > > **Weakness 5:** Theorem 6 seems confusing. In the prior section, you ... as well as the optimal ICL risks induced by them.
> >
> > **Response:**  Thank you for pointing out the potential confusion in Theorem 6. To clarify: the optimal WPGD risk $\mathcal L_{\texttt{WPGD}}^\star$ is defined in Eq. (3), where the search space for the weighting vector is $\omega\in\mathbb R^n$. In Theorem 5, we establish that $\mathcal L_{\texttt{GLA}}^\star=\mathcal L_{\texttt{WPGD}}^\star$ only when both Assumptions B and C hold. In contrast, Theorem 6 demonstrates that Assumption 2 alone is sufficient to achieve $\mathcal L_{\texttt{GLA-}v}^\star=\mathcal L_{\texttt{WPGD}}^\star$. We have revised the theorem statement to emphasize this distinction. We also added clarifications including the following discussion: “Under the bounded activation model of Assumption B, scalar gating is unable to express non-monotonic weighting schemes. For instance, suppose there are two tasks (T1 and T2): Even if T1 is more relevant to the query, Assumption B will result in assigning a higher weight to the examples in T2 which would lead to sub-optimal prediction. Theorem 6 shows that vector-valued gating can avoid such a bottleneck by encoding these tasks in distinct subspaces thanks to its enhanced expressivity.”
> >
> > On a related note, since vector gating can implement coordinate-wise weighting, it has the potential to achieve further improvements when the coordinates of the feature vector $x$ are not i.i.d. We believe this is an intriguing topic and propose it as a promising direction for future research.
> >
> > ---
> >
> > Due to space constraints, some clarifications and discussions were condensed in the original submission. We hope the revisions and additional explanations provided address the reviewer’s concerns. We believe that our work offers substantial contributions (as detailed in General Response) and would greatly appreciate it if the reviewer could reconsider their evaluation. We are happy to engage in further discussion to address any remaining questions or concerns.

---

### Official Review · Reviewer_N2zo · 2024-11-04

**Soundness:** 2
**Presentation:** 2
**Contribution:** 2
**Rating:** 5
**Confidence:** 3

**Summary:**

The authors study In-Context-Learning (ICL) abilities of a broad and general class of recent sequence-to-sequence architectures, such as Mamba, RWKV, and Gated Linear Attention, which they collectively name GLA. They show that the models of this algorithm family with L layers can perform L steps of Weighted Projected Gradient Descent (WPGD) for the task of linear regression during forward pass when presented with several examples. Moreover, the authors examine the setting when multi-task ICL examples are drawn from different distributions and correlated with the target sample. They find out, both empirically and theoretically, that for several weighting mechanisms,  it is possible to reach the optimal loss in this task.

**Strengths:**

1. Authors encompass different and seemingly distinct model families such as Mamba and Linear Transformer into one analytical framework and study their ICL properties jointly which is a very convenient and refreshing paradigm.
2. If the results stated in the paper are correct, it would be exciting and reassuring to know that a very broad class of models, beside Transformers, is provably capable of In-Context Learning despite some findings in the literature (e. g. [1])

[1] Jelassi, Samy, et al. "Repeat after me: Transformers are better than state space models at copying."

**Weaknesses:**

Unfortunately, I was not able to fully verify the preconditions and mathematical derivations for the results in the paper. Below there are several examples of what I found confusing:

1. Presentation is not self-contained, and it’s difficult to follow the arguments without prior reading of several referenced papers, notably Von Oswald et al. (2023), Ahn et al. (2024), Li et al. (2024), as well as  attempting to independently draw missing analogies and reductions between them and the reviewed paper.
2. In the papers (Ahn et al. (2024), Li et al. (2024)) mentioned above, their authors research Preconditioned Gradient Descent. In the reviewed paper, the authors discuss (Weighted) Projected Gradient Descent, although it’s ostensibly implied in line 68 to be the same algorithm. It’s unclear whether it’s indeed the same algorithm, and what are the differences between them if it’s not. It’s worth noting that the terms “projected GD” and “preconditioned GD” refer to different algorithms, in case the authors use them interchangeably.
3. If I understand correctly, the equation 7 states that parameter **$\hat{\beta}$** is the resulting predictor of data-dependent WPGD algorithm. **$\hat{\beta}$**  is constructed as a function of parameters $P_1, P_2$ and $\Omega$. However, formal definition of the algorithm, its connection to the parameters, and derivation on how it optimizes linear regression task and arrives at solution **$\hat{\beta}$** conditioned on priors $P_1, P_2$ and $\Omega$,  are not provided.
4. Moreover, The theorem 1 states that  the output of single-layer GLA model with a specific construction of weights matrices parametrized by $P_1, P_2$ and $\Omega$  matches the prediction of **$\hat{\beta}$** , also specifically constructed and parametrized by $P_1, P_2$ and $\Omega$.  This joint construction does not shed light on whether the forward pass of 1-layer GLA performs one step of gradient descent or WGPD. It would be helpful to explicitly demonstrate and prove it as in e.g. Lemma 1 of Ahn et al. (2024).

References:

Kwangjun Ahn, Xiang Cheng, Hadi Daneshmand, and Suvrit Sra. Transformers learn to implement preconditioned gradient descent for in-context learning. Advances in Neural Information Processing Systems, 36, 2024.

Yingcong Li, Ankit Singh Rawat, and Samet Oymak. Fine-grained analysis of in-context linear estimation: Data, architecture, and beyond. arXiv preprint arXiv:2407.10005, 2024.

Johannes Von Oswald, Eyvind Niklasson, Ettore Randazzo, João Sacramento, Alexander Mordvintsev, Andrey Zhmoginov, and Max Vladymyrov. Transformers learn in-context by gradient descent. In International Conference on Machine Learning, pp. 35151–35174. PMLR, 2023.

**Questions:**

See weaknesses.

Also, there might be some typos or minor mistakes that I would consider clarifying or correcting:

Line 158: $Z^\top Z$ instead of $Z Z^\top$ in formula for $\hat{y}$.

Line 161: It seems “predictor B” is associated with linear regression, not with linear attention.

Line 162: Perhaps, the authors meant one step **of** gradient descent?

Lines 202, 771, 775 and other throughout the paper: the authors interchangeably use bold **0** both as vector and a scalar, occasionally in the same line, which is confusing.

---

> ### Author Response · Authors · 2024-11-23
>
> > **Strength 2:** If the results stated in the paper are correct, it would be exciting and reassuring to know that a very broad class of models, beside Transformers, is provably capable of ICL despite some findings in the literature (e.g. [1])
>
> **Response:** Thank you for recognizing the strengths of our paper. Our framework builds on the observation that the Mamba architecture leverages a gated linear attention layer. Notably, our findings align with those of Jelassi et al. [1], which highlight the limitations of recurrent models in memory-intensive ICL tasks like associative recall. In contrast, we focus on characterizing algorithms expressible through gated linear attention, particularly for linear regression-type ICL tasks. Additionally, by distinguishing between "gated linear attention and linear attention" and "scalar-valued and vector-valued gating," our results shed light on how more advanced gating mechanisms can more effectively use the memory to enhance recall capabilities.
>
> > **Weakness 1:** Presentation is not self-contained, and it’s difficult to follow the arguments without prior reading of several referenced papers,... as well as attempting to independently draw missing analogies and reductions between them and the reviewed paper.
>
> **Response:**   We appreciate the reviewer’s feedback on the presentation of our paper. In response, we have revised Section 3 to provide a clearer explanation of the connection between GLA and WPGD, thereby better motivating our contributions. Additionally, we have outlined the novelty and key contributions of our work in General Response. We welcome further suggestions from the reviewer if there are any remaining points that require clarification.
>
> > **Weakness 2:** In the papers (Ahn et al. (2024), Li et al. (2024)) mentioned above, their authors research Preconditioned Gradient Descent. ... in case the authors use them interchangeably.
>
> **Response:** Thank you for highlighting the confusion regarding terminology. We used the term "Projected Gradient Descent" because the $P_1$ and $P_2$ matrices in Eq. (7) can be viewed as projection matrices. However, we agree with the reviewer that "Preconditioned Gradient Descent" is more accurate and aligns better with the terminology used in the referenced literature. We have updated the paper to adopt this terminology consistently throughout.
>
> > **Weakness 3:** If I understand correctly, the equation 7 states that ... on priors $P_1$, $P_2$, and $\Omega$, are not provided.
>
> **Response:**  Thank you for your question.  We have enhanced the exposition for clarity. In Section 3.1, we now begin with a standard “weighted least-squares objective” to first derive the scalar-weighted PGD algorithm. This corresponds to scalar-gated linear attention. We then generalize this to the vector-weighted PGD estimator $\hat{\beta} = \beta_1^{\text{gd}}(P_1, P_2, \Omega)$, as defined in Eq. (7), with preconditioning matrices $P_1,P_2$ and weighting matrix $\Omega$.
>
> Our Theorem 1 establishes the mapping between this vector-weighted PGD and the attention weights in Eq. (8) and weights induced by vector-valued gating.  To reduce ambiguity, we have revised Theorem 1 to include additional explanations that explicitly clarify the connection between GLA and WPGD, making it easier to understand.
>
> > **Weakness 4:** Moreover, Theorem 1 states that ... It would be helpful to explicitly demonstrate and prove it as in e.g., Lemma 1 of Ahn et al. (2024).
>
> **Response:** Thank you for your comment. Our Theorem 2, which extends to multilayer architectures, builds directly upon and generalizes Lemma 1 of Ahn et al. (2024). It addresses additional complexities introduced by causal masking and non-linear gating mechanisms, which are crucial and widely used in practical applications but were not considered in previous theoretical work.
>
> Additionally, we recently came across the work by [Ding et al. (2024)](https://arxiv.org/pdf/2308.06912), which provides a theoretical analysis of causal masking in in-context learning. Their findings demonstrate that causal language models (causalLM) exhibit suboptimal convergence dynamics akin to those of online gradient descent with non-decaying step sizes. This behavior limits their ability to reach optimal solutions, even with an increasing number of in-context examples. We have cited this work below Theorem 2 to further underscore the challenges posed by gated and causally-masked architectures, which justify the extensions and contributions.
>
> To further clarify, we have reorganized Theorem 1 and Section 3.1 to explicitly clarify that the forward pass of a one-layer GLA corresponds to a one step of WPGD.
>
> > **Questions:** Also, there might be some typos or minor mistakes that I would consider clarifying or correcting.
>
> **Response:** Fixed. Thank you!

---

> > ### Author Response · Authors · 2024-11-29
> >
> > Dear Reviewer, as the discussion phase is coming to an end, we would be grateful to hear if you have further feedback.

---

> ### Comment · Reviewer_N2zo · 2024-12-03
>
> Dear authors,
>
> Thank you for your response and sorry for the late answer.
>
> I acknowledge that revision of the paper made it more clear.  However, I remain skeptical due to the following:
>
> $\beta_1^{\text{gd}}(P_1, P_2, \Omega) := P_2 \big( X P_1 \odot \Omega \big)^\top y$
>
> is not a preconditioned gradient descent as it should have only one preconditioner matrix at the leftmost side of the expression as in the formula in line 178. For reference, see definitions of preconditioned gradient descent in e.g. https://www.cs.cornell.edu/courses/cs4787/2019sp/notes/lecture8.pdf, https://www.cs.princeton.edu/courses/archive/fall18/cos597G/lecnotes/lecture5.pdf, and https://www.cs.princeton.edu/%7Earora/TheoryDL.pdf (2.4.1). It seems that you creatively introduced an additional parameter $P_1$ to the PGD formula so it could align well with your derivation for the GLA output.
>
> It stands as a major problem for me, because I believe it makes the core claim of the paper in its current state that the GLA implements WPGD unproved.
>
>
> And, anticipating a possible argument that $P_1$ could be treated as a parameter of underlying regression algorithm rather than a second "preconditioner", I note that in such a case 1) the underlying algorithm would likely no longer be an ordinary linear regression; 2) As an optimizable parameter in the corresponding GLA layer, it would also be optimizable in a regression, and the gradient would have been taken w.r.t both $\beta$ and $P_1$ further complicating matters.
>
>
> Additionally, I would like to acknowledge that SSMs such as Mamba and RNNs (RWKV) are fundamentally different from Linear Transformers in their core sequence mixing mechanism, despite the similarities in gating mechanisms which were discussed in the GLA paper [1]. One class of models cannot be readily re-parametrized to represent another. Your derivations and proofs are valid only in case of the eponymous Gated Linear Attention class of models from the paper [1]. Therefore, I strongly suggest to remove the mentions of various recurrent and state space models which could mislead the reader into believing your results cover those models too.

---

> > ### Author Response · Authors · 2024-12-04
> >
> > We thank the reviewer for their thoughtful feedback and additional questions. We aim to address the raised concerns below:
> > > I acknowledge that revision of the paper made it more clear. … the gradient would have been taken w.r.t both $\beta$ and $P_1$ further complicating matters.
> >
> > **Response:** Our definition in Eq. (7a) is a strict generalization of weighted PGD (WPGD). **Kindly note that it exactly matches the reviewer’s definition of WPGD when the gating function is scalar or vector-based**, as discussed in Section 3.3. We elaborate on different scenarios further below.
> >
> > * **First, when the gating function is constant**, let the gating function be $\Omega = c11^\top$, where $c$ is a nonzero constant. In this case, our method reduces to the standard PGD from earlier works$$\beta_1^{\text{gd}}(P_1, P_2, c) = c (P_2P_1^\top)X^\top y.$$
> > Here, $P=P_2P_1^\top$ acts as the preconditioner, and $ c X^\top y$ represents the gradient of the least squares objective multiplied by $c$  i.e., $\mathcal{L}(\beta) = c \sum_{i=1}^n \cdot \left( y_i - \beta^\top x_i \right)^2$ at $\beta=\beta_0=0$.  **This matches the classical definition of PGD** as the reviewer referenced and is widely used in previous in-context learning literature on PGD [[Ahn et. al. 2023](https://arxiv.org/pdf/2306.00297)].
> >
> > * **Second, when the gating function is scalar- or vector-based**, let $G_i = \gamma_i 11^\top$ for scalar gating and $G_i = \alpha_i 1^\top$ for vector gating, where $\gamma_i \in \mathbb{R}$ and $\alpha_i \in \mathbb{R}^{d+1}$ (as discussed in Section 3.3). The corresponding weighting matrix $\Omega$ in Theorem 1 simplifies to $\Omega=\omega 1^\top$ where
> > $\omega = [\gamma_i\cdots\gamma_{n}]^\top \in \mathbb{R}^n$ for scalar gating and $\omega = [\alpha_{1,d+1}\cdots \alpha_{n,d+1}]^\top \in \mathbb{R}^n$ for vector gating.
> > Substituting into the update, we obtain $$\beta_1^{\text{gd}}(P_1, P_2, \Omega) = P_2(XP_1 \circ \Omega)^\top y = P_2P_1^\top X^\top (y\circ\omega).$$
> > In this case, $P=P_2P_1^\top$ is the preconditioner, and $X^\top (y \circ \omega)$ is the gradient of the weighted least squares objective at $\beta_0=0$. **This aligns with the definition of weighted PGD (WPGD) [[Li et. al 24](https://arxiv.org/pdf/2407.10005)]**.
> >
> > * **Finally, when the gating function is matrix-based**, $\Omega$ is a matrix. This corresponds to Eq. (7a) in the paper:$$\beta_1^{\text{gd}}(P_1, P_2, \Omega) = P_2(XP_1\circ\Omega)^\top y.$$
> > Unlike the previous cases, the preconditioning matrices $P_1$ and $P_2$ **cannot** collapse into a single preconditioner $P = P_2P_1^\top$ due to the coordinate-wise weighting introduced by $\Omega$. **This scenario corresponds to our “strict generalization of WPGD”**, which applies coordinate-wise weighting, allowing for greater flexibility in adapting to the structure of the data.
> >
> > Overall, “data-dependent WPGD” captures the core essence of the algorithm. As the gating function becomes more sophisticated, the algorithm transitions progressively from PGD to WPGD and ultimately to a “strict generalization of WPGD.” The phrase “general class of WPGD algorithms with data-dependent weights” in the abstract is intended to convey this progression.
> > > Additionally, I would like to acknowledge that SSMs such as Mamba and RNNs (RWKV) are fundamentally different from Linear Transformers ... which could mislead the reader into believing your results cover those models too.
> >
> > **Response:** Thanks for bringing this up. We are not claiming that Mamba and RWKV-6 are exactly identical to GLA. Instead, they are using the same core recurrence mechanism of GLA, and thus can be viewed as variations of GLA. For instance, selective state-space models like Mamba use time-varying state space parameterization with $(A_t,B_t,C_t)$ matrices. In Mamba and Mamba-2, the authors choose $B_t$ and $C_t$ matrices as linear functions of the token $x_t$ (e.g., at the bottom of Page 5 of [Mamba](https://arxiv.org/pdf/2312.00752) and Page 26 of [Mamba-2](https://arxiv.org/pdf/2405.21060)). With this choice, time-varying SSM directly corresponds to gated linear attention where $(B_t,C_t,x_t)$ play the roles of $(k_t,q_t,v_t)$ and the state matrix $A_t$ corresponds to the gating scheme. Specifically, for $A_t = \omega_tI$ (as in Dao & Gu, 2024), we derive:
> > $$h_t=A_th_{t-1}+B_tx_t=\omega_th_{t-1}+v_tq_t^\top\quad\text{and}\quad o_t=C_t^\top h_t=q_t^\top h_t,$$
> > which matches the recurrent form of GLA in Eq. (1).  Similarly, RWKV-6 (e.g., Section 4 in [Peng et al., 2024](https://arxiv.org/pdf/2404.05892)) also employs a recurrent form that aligns closely with GLA’s formulation.
> > Additional discussion can also be deduced from the Mamba-2 paper which makes explicit connections between the linear transformers and SSMs. The [GLA paper](https://arxiv.org/pdf/2312.06635) (as well as xLSTM and RWKV-6 paper) also makes similar connections/choices.
> >
> > ---
> > We understand reviewer response time is done and sincerely hope our response addresses their concerns.

---

### Official Review · Reviewer_Kuke · 2024-11-08

**Soundness:** 3
**Presentation:** 2
**Contribution:** 2
**Rating:** 6
**Confidence:** 3

**Summary:**

This paper shows that Gated Linear Attention (GLA) can implement Weighted Projected Gradient Descent (WPGD) algorithms. Furthermore, the gating mechanism in GLA allows the in-context samples to come from distinct tasks. This paper also characterizes the loss landscape of WPGD and one-layer GLA.

**Strengths:**

1. This paper establish the equivalence between GLA and WPGD. The authors show that GLA can weight the context window through gating, so GLA can learn from non-IID in-context samples, while linear attention can't.
2. This paper characterizes the loss landscape of GLA and WPGD and shows that the WPGD minimizer is unique.

**Weaknesses:**

1. The attention matrices are restricted in certain form (eq (8), line 201-203). In actual training setting of GLA, the learned attention matrices may not have such forms. It's not sure whether GLA implement WPGD when attention matrices don't have these restricted forms.

2. The token embeddings also have restricted forms (line 426). It is not clear whether GLA can learn the contextual features if the token embeddings are learnable parameters.

**Questions:**

1. Line 486 says that Assumption B ensures that any $\omega$ in W can be achieved by an appropriate choice of gating parameters. i'm not sure this statement is correct. If the number of tasks $K$ is larger than the dimension of $w_g$ (the trainable parameter of the gating function), the above statement seems to be wrong.

2. Theorem 2 states that an L-layer GLA implements L steps of WPGD. The question is when L is large enough, whether the L-layer GLA found the better predictor than the one-layer GLA. Can Theorem 2 show the advantage of the deeper models?

3. Are there more experiments of multi-layer GLA?

---

> ### Author Response · Authors · 2024-11-23
>
> > **Weakness 1:** The attention matrices are restricted in certain form (eq (8), line 201-203). ... It's not sure whether GLA implements WPGD when attention matrices don't have these restricted forms.
>
> **Response:** Thank you for pointing this out. We would like to emphasize that the restricted form/construction of attention matrices in Eq. (8) is not an arbitrary choice but is well-supported by prior literature. Several existing works have adopted similar constructions:
>
> 1. Proposition 1 in [Von Oswald et al. (2023)](https://arxiv.org/pdf/2212.07677), Section 2.4 in [Lu et al. (2024)](https://arxiv.org/pdf/2405.11751), and Appendix B in [We et al. (2024)](https://arxiv.org/pdf/2310.08391) all make similar assumptions about attention weights.
> 2. Theorem 1 in [Ahn et al. (2024)](https://arxiv.org/pdf/2306.00297) and Proposition 1 in [Li et al. (2024)](https://arxiv.org/pdf/2407.10005) demonstrate that optimizing single-layer linear attention models with or without such restrictions yields equivalent predictions.
> 3. Theorem 4.1 in [Zhang et al. (2024)](https://arxiv.org/pdf/2306.09927) and Eq (2) in [Huang et al. (2023)](https://arxiv.org/pdf/2310.05249) show that, when attention weights are initialized following similar constraints, their structural form persists throughout training, as zero entries remain zero and weights converge to forms consistent with Eq. (8).
>
> We note that our study focuses on the GLA model with a nonlinear gating function and task mixtures where sequences include **multiple tasks**. These settings introduce significantly more complexity compared to prior work, even under the constrained attention weight forms discussed in Eq. (8). Hence, our contributions extend beyond existing findings on single-task ICL and are non-trivial.
>
> > **Weakness 2:** The token embeddings also have restricted forms (line 426). It is not clear whether GLA can learn the contextual features if the token embeddings are learnable parameters.
>
> **Response:** In both this work and previous studies on linear attention architectures, $z$ in $Z$ is referred to as both input tokens and embeddings due to the linearity of the model. If we consider a learnable linear embedding matrix $W_e$, the prediction for linear attention can be expressed as:
> $$
> \text{LinAtt}(Z) = (ZW_eW_qW_k^{\top} (ZW_e)^\top)ZW_eW_v=
> (Z(W_eW_qW_k^{\top} W_e^\top)Z^\top)Z(W_eW_v) = (Z(W_q'W_k'^{\top})Z^\top)ZW_v',
> $$
> where $W_{q,k,v}'=W_eW_{q,k,v}$.
> Thus, the embedding matrix $W_e$ can be absorbed into the attention weights, yielding equivalent optimization results. This is why our framework does not consider learnable token embeddings without loss of generality.
>
> Additionally, in our setting, the contextual features in Eq. (20) can be arbitrary random vectors once they are linearly independent (as per Assumption B). Therefore, this setup is broad and highly general.
>
> > **Question 1:** Line 486 says that Assumption B ensures that any $\omega$ in $W$ can be achieved by an appropriate choice of gating parameters. ... If the number of tasks $K$ is larger than the dimension of $w_g$ (the trainable parameter of the gating function), the above statement seems to be wrong.
>
> **Response:** You are correct that if the number of tasks $K$ exceeds the dimension of $w_g$, not all $\omega \in \mathcal W$ can be represented via the gating parameters. However, Assumption B explicitly requires that the delimiters be linearly independent, which already implies that $K <\dim(w_g)$, as a larger $K$ would violate the linear independence condition and render Assumption B invalid.
>
>
> > **Question 2:** Theorem 2 states that an $L$-layer GLA implements $L$ steps of WPGD. The question is: when $L$ is large enough, does the $L$-layer GLA find a better predictor than the one-layer GLA? Can Theorem 2 demonstrate the advantage of deeper models?
>
>
> **Response:**  It is well-established that additional steps of gradient descent (with appropriately chosen step sizes) generally result in reduced loss. In an $L$-layer GLA, each layer corresponds to a step of GD as outlined in Theorem 2, meaning the $L$-layer GLA effectively performs $L$ steps of GD. Consequently, it achieves improved predictions compared to a single-layer GLA for $L > 1$.
>
> > **Question 3:** Are there more experiments of multi-layer GLA?
>
> **Response:**  In response to the reviewer’s request for additional experiments, we have added results in Appendix A.1 of the revised submission. Due to time constraints, we replicated the setting from Fig. 1(a) to demonstrate the improvements provided by deeper models. The additional experimental results again verify that deeper models yield better predictions. We are also considering conducting further experiments in different settings for the final submission.

---

> > ### Author Response · Authors · 2024-11-30
> >
> > Dear Reviewer, as the discussion phase is coming to an end, we would be grateful to hear if you have further feedback.

---

> > > ### Comment · Reviewer_Kuke · 2024-12-03
> > >
> > > Thank you for your response. My score remains unchanged.

---

### Author Response · Authors · 2024-11-23
**General Response**

We thank all the reviewers for their thoughtful feedback and insightful comments, which have been invaluable in improving the manuscript. Below, we highlight the main contributions (**C1-C3**) of the paper, summarize the main points (**P1-P4**) raised by the reviewers, and provide an overview of our actions (**A1-A4**).

---
 **C1. Bridging GLA and WPGD:** We establish a rigorous connection between Gated Linear Attention (GLA) and data-dependent Weighted Preconditioned Gradient Descent (WPGD) in **multi-task** in-context learning (ICL), showing that gating mechanisms enable dynamic task weighting beyond linear attention’s static behavior. Reviewer Kuke remarked: "This paper establishes the equivalence between GLA and WPGD. GLA can weight the context window through gating, so GLA can learn from non-IID in-context samples, while linear attention can't."

 **C2. Global Optimization Landscape Analysis of WPGD/GLA:** This paper provides the **first** comprehensive analysis of the global optimization landscape of GLA in ICL. Specifically, we characterize its loss landscape and, as demonstrated in Theorem 4, show that under mild conditions, there exists a unique global minimum up to scaling invariance. We believe Theorem 4 fills a critical gap in understanding the global optimization landscape of attention mechanisms, a contribution that may not have been fully recognized in the reviewers' feedback.

 **C3. Novel Theoretical Insights:** We develop innovative tools for analyzing GLA's optimization geometry, rigorously investigating how task correlations influence convergence and **comparing scalar- and vector-gated mechanisms**. Reviewer 41dD noted: "They also have experimental evidence to show the performance gap among scaler-gated, vector-gated, and vanilla linear self-attention." Reviewer Y6dU added: "Comparing the effects of scalar and vector gating mechanisms on performance provides valuable insights for crafting models."

---
We summarize the main points raised by the reviewers:


**P1. Clarification of Assumptions (Reviewers Kuke and 41dD):** The need to justify assumptions, such as the attention weight construction, independence of delimiters and task correlations, was highlighted, along with their impact on the theoretical results.

**P2. Paper Presentation (Reviewers N2zo and 41dD):** Clarifications were requested regarding the equivalence claims between GLA and WPGD, as well as the connections between scalar- and vector-gated attention mechanisms.

**P3. Empirical Validation (Reviewers Kuke, 41dD, Y6dU, and xUCq):** Reviewers suggested additional experiments to validate theoretical results and assess the practical applicability of GLA, including the role of contextual features and delimiters.

**P4. Broader Context and Related Work (Reviewer Y6dU):** The need to strengthen the connections to related works.

---
We have taken several significant steps to address these concerns:

**A1**: We clarified the theoretical contributions and the role of assumptions. These changes include reorganizing Theorem 1, adding Corollary 4, and updating the Related Work section.

     Further details are provided in the response to Reviewers Kuke and 41dD.

**A2**: We revised the manuscript to improve the presentation of key concepts, ensuring that the paper is more self-contained. Specific changes were made to Sections 3 and 5 to address ambiguities and improve clarity.

     Further details are provided in the response to Reviewer N2zo and 41dD.

**A3**: We introduced new multi-layer experiments in Appendix A.1, and further discussions on real-data applications in Section 5.

     Further details are provided in the response to Reviewer Kuke, 41dD, Y6dU, and xUCq.

**A4**:  We have included discussions in Section 1.1, linking GLA to (unified) implicit attention frameworks and gated cross-attention models.

     Further details are provided in the response to Reviewer Y6dU.


The revised text in the manuscript is highlighted in blue for clarity. Further details are provided in the responses to individual reviewers. We believe these revisions substantially address the reviewers' comments, and we look forward to receiving any additional feedback.

---

### Author Response · Authors · 2024-11-28
**Happy to Address Any Further Concerns**

Dear Reviewers,

We sincerely appreciate the time and effort you have dedicated to reviewing our work.

As the discussion period and revision deadline approach, we would greatly appreciate any additional feedback to ensure we have addressed all your questions and concerns.

Best regards,

The Authors

---

### Author Response · Authors · 2024-12-04
**Thanks to the Reviewers**

We would like to thank all the reviewers for their constructive comments, which have greatly helped to improve both the clarity and content of the paper.

Best,

The Authors

---

### Meta-Review · Area_Chair_Y1gD · 2024-12-20

**Metareview:**

The paper aims to show that Gated Linear Attention can be interpreted as implementing Weighted Projected Gradient Descent in in-context learning scenarios.

Despite the authors’ detailed rebuttals and clarifications, the major concern from Reviewer N2zo during the Reviewer-AC discussion remain unaddressed to a satisfactory extent. The fundamental claim that GLA straightforwardly implements a known form of PGD is called into question, and the literature or theoretical arguments do not convincingly validate this new “generalized preconditioned gradient descent” concept. Adding the matrix P in preconditioned gradient descent has a well-established explanation, but after generalization, it is unclear whether it still qualifies as gradient descent or merely resembles it.

For the benefit of this paper, we regretfully reject it for now. Note that this is not a discouragement, but rather an encouragement for the authors to make use of the reviewers' comments to add more clarification, improve the work, and achieve broader impact. We believe this paper has the potential to become a strong submission in the future.

**Additional Comments On Reviewer Discussion:**

During the final stages of the discussion, Reviewer N2zo reiterated critical objections that remained unresolved despite the authors’ rebuttal:

- **Incorrect Claims About PGD/WPGD Equivalence:**
  Reviewer N2zo strongly contested the core claim that GLA implements WPGD. While the authors attempted to frame certain update formulas as forms of “generalized preconditioned gradient descent,” the reviewer argued that these steps deviate from standard definitions of preconditioned or projected gradient descent. the reviewer noted that a proper PGD update rule typically involves a single well-defined preconditioner matrix, whereas the authors’ proposed formulation included multiple matrices, making it hard to interpret as a standard PGD-based method. This is a major concern which fundamentally challenges the paper’s main point of GLA implementing a form of gradient descent for linear regression.



- **Incorrect Statements About Models Like Mamba and RWKV:**
Additionally Reviewer N2zo also disputed the authors’ suggestions that models like Mamba (a state-space model) can be straightforwardly encompassed by the GLA framework. The reviewer emphasized that sequence mixing mechanisms in SSM-based architectures are fundamentally different. The Reviewer N2zo provided 2 papers [1,2], which show that Mamba and other SSMs underperform Transformers on real and synthetic In-Context-Learning tasks.

[1] Jelassi, S., Brandfonbrener, D., Kakade, S.M. & Malach, E.. (2024). Repeat After Me: Transformers are Better than State Space Models at Copying. <i>Proceedings of the 41st International Conference on Machine Learning</i>, in <i>Proceedings of Machine Learning Research</i> 235:21502-21521 Available from https://proceedings.mlr.press/v235/jelassi24a.html.

[2] Waleffe, R., Byeon, W., Riach, D., Norick, B., Korthikanti, V.A., Dao, T., Gu, A., Hatamizadeh, A., Singh, S., Narayanan, D., Kulshreshtha, G., Singh, V., Casper, J., Kautz, J., Shoeybi, M., & Catanzaro, B. (2024). An Empirical Study of Mamba-based Language Models. ArXiv, abs/2406.07887. Available from https://arxiv.org/abs/2406.07887v1

---

### Decision · Program_Chairs · 2025-01-22

Reject